# Translational initiation factor eIF5 replaces eIF1 on the 40S ribosomal subunit to promote start-codon recognition

Jose Luis Llácer[1,2†‡], Tanweer Hussain[3†], Adesh K Saini[4†], Jagpreet Singh Nanda[5†], Sukhvir Kaur[4], Yuliya Gordiyenko[1], Rakesh Kumar[4], Alan G Hinnebusch[6*], Jon R Lorsch[5*], V Ramakrishnan[1*]

[1]MRC Laboratory of Molecular Biology, Cambridge, United Kingdom; [2]Instituto de Biomedicina de Valencia (IBV-CSIC), Valencia, Spain; [3]Department of Molecular Reproduction, Development and Genetics, Indian Institute of Science, Bangalore, India; [4]Shoolini University of Biotechnology and Management Sciences, Himachal Pradesh, India; [5]Laboratory on the Mechanism and Regulation of Protein Synthesis, Eunice K Shriver National Institute of Child Health and Human Development, National Institutes of Health, Bethesda, United States; [6]Laboratory of Gene Regulation and Development, Eunice K Shriver National Institute of Child Health and Human Development, National Institutes of Health, Bethesda, United States

*For correspondence:
alanh@mail.nih.gov (AGH);
jon.lorsch@nih.gov (JRL);
ramak@mrc-lmb.cam.ac.uk (VR)

†These authors contributed equally to this work

Present address: ‡Instituto de Biomedicina de Valencia (IBV-CSIC), Valencia, Spain

**Abstract** In eukaryotic translation initiation, AUG recognition of the mRNA requires accommodation of Met-tRNA$_i$ in a 'P$_{IN}$' state, which is antagonized by the factor eIF1. eIF5 is a GTPase activating protein (GAP) of eIF2 that additionally promotes stringent AUG selection, but the molecular basis of its dual function was unknown. We present a cryo-electron microscopy (cryo-EM) reconstruction of a yeast 48S pre-initiation complex (PIC), at an overall resolution of 3.0 Å, featuring the N-terminal domain (NTD) of eIF5 bound to the 40S subunit at the location vacated by eIF1. eIF5 interacts with and allows a more accommodated orientation of Met-tRNA$_i$. Substitutions of eIF5 residues involved in the eIF5-NTD/tRNA$_i$ interaction influenced initiation at near-cognate UUG codons in vivo, and the closed/open PIC conformation in vitro, consistent with direct stabilization of the codon:anticodon duplex by the wild-type eIF5-NTD. The present structure reveals the basis for a key role of eIF5 in start-codon selection.
DOI: https://doi.org/10.7554/eLife.39273.001

## Introduction

Eukaryotic translation initiation is a multistep process that involves assembly of a pre-initiation complex (PIC) comprised of the small (40S) ribosomal subunit, methionyl initiator tRNA (Met-tRNA$_i$) and numerous eukaryotic initiation factors (eIFs). The binding of this 43S PIC to the capped 5' end of mRNA is followed by scanning the mRNA leader for the correct AUG start codon. The binding of eIF1 and eIF1A to the 40S subunit promotes a scanning-conducive, open conformation favourable for rapid binding of Met-tRNA$_i$ as a ternary complex (TC) with eIF2-GTP, in a conformation (P$_{OUT}$) suitable for scanning successive triplets in the 40S P site for complementarity to the anticodon of Met-tRNA$_i$. The multisubunit eIF3 complex also binds directly to the 40S subunit and stimulates 43S assembly, attachment to mRNA, and subsequent scanning. During the scanning process, hydrolysis of GTP in TC is stimulated by the GTPase activating protein (GAP) eIF5, but release of phosphate (P$_i$) from eIF2-GDP-P$_i$ is prevented by the gatekeeper molecule eIF1 at non-AUG codons.

Recognition of an AUG start codon induces a major conformational change in the PIC to a scanning-arrested closed (P$_{IN}$) complex made possible by the dissociation of eIF1, which eliminates a clash it would have with Met-tRNA$_i$ in its fully accommodated P$_{IN}$ conformation. The change is accompanied by the movement of the C-terminal tail (CTT) of eIF1A from eIF1 toward the GAP domain of eIF5. The P$_{IN}$ conformation is further stabilized by direct interaction of the unstructured N-terminal tail (NTT) of eIF1A with the codon-anticodon duplex (*Hinnebusch, 2014*; *Hinnebusch, 2017*; *Aylett and Ban, 2017*).

In addition to its function as a GAP for eIF2, eIF5 has been implicated in stringent selection of AUG start codons. eIF5 consists of an N-terminal domain (NTD, 1–170 residues), which is connected by a long flexible linker to a C-terminal domain (CTD) (*Conte et al., 2006*; *Wei et al., 2006*). The eIF5-NTD contains the GAP function of eIF5, which is stimulated by the PIC and requires the conserved Arg-15 located in its unstructured N-terminus, possibly functioning as an 'arginine finger' that interacts with the GTP-binding pocket in eIF2γ to stabilize the transition state for GTP hydrolysis (*Algire et al., 2005*; *Das et al., 2001*; *Paulin et al., 2001*). Substitution of Gly31 with arginine in the eIF5-NTD (the *SUI5* allele) is lethal but confers a dominant Sui⁻ (*su*ppressor of *i*nitiation codon) phenotype indicating error-prone start-codon selection in yeast cells (*Huang et al., 1997*). Consistent with this, in the reconstituted system, the eIF5-G31R substitution alters the regulation of P$_i$ release such that it occurs faster at UUG versus AUG start codons; and it modifies a functional interaction of eIF5 with the eIF1A-CTT to favour the closed PIC conformation at UUG over AUG start codons. Biochemical analysis of intragenic suppressors of *SUI5* with an Ssu⁻ (*Suppressor of Sui*⁻) phenotype, indicating hyperaccurate start-codon selection, indicated that the effect of eIF5-G31R in both dysregulation of P$_i$ release and partitioning of PICs between open and closed states contribute to the enhanced UUG initiation *in vivo* (*Maag et al., 2006*; *Saini et al., 2014*). Because the location of eIF5 in the PIC was unknown, it has been unclear how the G31R substitution alters these events in the open/P$_{OUT}$ to closed/P$_{IN}$ transition at the molecular level. Movement of the wild-type eIF5-NTD and the eIF1A CTT toward one another within the PIC is triggered by AUG recognition, and this rearrangement is dependent on scanning enhancer (SE) elements in the eIF1A-CTT (*Nanda et al., 2013*). The fact that mutations in SE elements more strongly reduced the rate of P$_i$ release than eIF1 dissociation suggested that the SE-dependent movement of the eIF5-NTD toward the eIF1A-CTT facilitates P$_i$ release following eIF1 dissociation (*Nanda et al., 2013*).

The eIF5 CTD also performs multiple functions in assembly of the PIC and control of start-codon selection such as promoting the closed PIC conformation by enhancing eIF1 dissociation (*Nanda et al., 2009*; *Nanda et al., 2013*). Accordingly, overexpressing eIF5 in yeast (*Nanda et al., 2009*) or mammalian cells (*Loughran et al., 2012*) relaxes the stringency of start-codon selection presumably by enhancing eIF1 release (*Loughran et al., 2012*). Thus, the structure of a eukaryotic translation initiation complex containing eIF5 would greatly aid our understanding of its multiple functions.

The recent structures of eukaryotic translation initiation complexes from yeast (*Aylett et al., 2015*; *Hussain et al., 2014*; *Llácer et al., 2015*) as well as mammals (*Lomakin and Steitz, 2013*; *Aylett et al., 2015*; *Hashem et al., 2013*) have provided many insights into the molecular events involved in scanning and AUG recognition. eIF1, eIF1A, TC and eIF3 have been observed in an open (P$_{OUT}$) conformation, as well as a scanning-arrested closed (P$_{IN}$) conformation of the 40S (*Llácer et al., 2015*). In a partial 48S pre-initiation complex from yeast (py48S), we had tentatively suggested that an unassigned density at low resolution near eIF2γ belongs to eIF5-CTD (*Hussain et al., 2014*). However, there is no clear structural information on the position and conformation of either the CTD or NTD of eIF5 in the PIC.

Here, we have determined a cryo-EM structure of a yeast 48S complex in the P$_{IN}$ conformation at near atomic resolutions (3.0 Å to 3.5 Å maps) containing clear density for the eIF5-NTD (py48S-eIF5N). Remarkably, in this py48S-eIF5N complex, eIF1 has been replaced by the eIF5-NTD, which is bound near the P site at essentially the same position. The tRNA$_i$ is more fully accommodated in the P site than observed in previous structures containing eIF1, and is also tilted toward the 40S body, apparently setting the stage for its interaction with eIF5B and subsequent joining of the 60S subunit. Extensive interaction with the eIF5-NTD appears to stabilize this tRNA$_i$ conformation. Mutations expected to weaken the observed eIF5-NTD/tRNA$_i$ interactions diminish initiation at near-cognate UUG codons *in vivo* and disfavor transition to the closed/P$_{IN}$ conformation at UUG codons in reconstituted PICs *in vitro*, whereas mutations expected to stabilize the interactions have the opposite

effects. The results presented in this work suggest that the eIF5-NTD stabilizes the codon-anticodon interaction and the closed/P$_{IN}$ state of the PIC, prevents eIF1 rebinding, and promotes a conformation of the 48S PIC compatible with eIF5B binding and subunit joining.

## Results

### Overview of the cryo-EM structure of a yeast 48S PIC containing eIF5

Partial yeast 48S PIC maps containing clear density for eIF5-NTD (py48S-eIF5N) were obtained from a 48S sample reconstituted by sequential addition of purified *Saccharomyces cerevisiae* eIF1, eIF1A, eIF3, eIF5, TC and an eIF4F-eIF4B-mRNA complex to yeast *Kluyveromyces lactis* 40S subunits. An unstructured, capped 49-mer mRNA with an AUG codon and an optimal Kozak sequence for yeast was used (See Materials and methods). In our earlier studies, eIF4 factors were not used to deliver the mRNA to the PIC, and the mRNA was uncapped and lacked an optimal Kozak sequence. These changes in assembly protocol may have helped in capturing eIF5 in the 48S.

The structure of py48S-eIF5N was determined to an overall resolution of 3.0 Å to 3.5 Å in respective maps: 1, A, B, C1 and C2 (*Figure 1—figure supplements 1*, *2* and *3*; *Tables 1*, *2* and *3*), and the resulting model is shown in *Figure 1*. The local resolution of the density is highest for the 40S core and ligands directly attached to it, including the eIF5-NTD (*Figure 1—figure supplement 4* and *Table 2*). Met-tRNA$_i$ is bound to py48S-eIF5N in a P$_{IN}$ state (base paired to the AUG codon), with a closed mRNA latch and compressed conformation of h28 (the rRNA helix connecting the 40S

**Table 1.** Refinement and model statistics.

| | Model with TC in conformation 1 | Model with TC in conformation 2 |
|---|---|---|
| Model Composition | | |
| Non-hydrogen atoms | 104,332 | 104,232 |
| Protein residues | 8538 | 8522 |
| RNA bases | 1882 | 1882 |
| Refinement | | |
| Resolution used for refinement (Å) | 3.05 | 3.05 |
| Map sharpening B-factor (Å) | −67 | −66 |
| Average B-factor (Å) | 162 | 121 |
| Fourier Shell Correlation (FSC)* | 0.90 | 0.89 |
| Rms deviations | | |
| Bonds (Å) | 0.006 | 0.006 |
| Angles (°) | 1.139 | 1.194 |
| Validation (proteins) | | |
| Molprobity score (Percentile in brackets) | 2.65 (91st) | 2.74 (89th) |
| Clashscore, all atoms (Percentile in brackets) | 6.87 (100th) | 7.11 (98th) |
| Good rotamers (%) | 91.1 | 89.2 |
| Ramachandran plot | | |
| Favored (%) | 89.8 | 89.2 |
| Outliers (%) | 2.5 | 2.7 |
| Validation (RNA) | | |
| Correct sugar puckers(%) | 98.6 | 96.9 |
| Good backbone conformations(%) | 63.6 | 63.1 |

*FSC= $\Sigma$(Nshell FSCshell)/ $\Sigma$(Nshell), where FSCshell is the FSC in a given shell, Nshell is the number of 'structure factors' in the shell. FSCshell = $\Sigma$(Fmodel FEM)/ ($\sqrt{(\Sigma(|F|2model))}$ $\sqrt{(\Sigma(|F|2EM))}$)

DOI: https://doi.org/10.7554/eLife.39273.013

**Table 2.** Local resolution of ligands.

| Structures | Map 1 (Å) | Map A (Å) | Map B (Å) | Map C1 (Å) | Map C2 (Å) |
|---|---|---|---|---|---|
| Overall Resolution | 3.00 | 3.50 | 3.50 | 3.50 | 3.10 |
| eIF5-NTD | 3.15 | nd | nd | 3.80 | 3.35 |
| eIF1A | 3.00 | nd | nd | 3.70 | 3.10 |
| eIF2α | 3.65 | nd | nd | 4.15 | 3.65 |
| eIF2β | nd* | nd | nd | 9.40 | >15 |
| eIF2γ | 8.15 | nd | nd | 8.00 | 8.10 |
| tRNAi | 3.20 | nd | nd | 3.65 | 3.20 |
| ASL + mRNA(−4 to + 4) | 2.95 | nd | nd | 3.45 | 3.15 |
| eIF3 PCI domains | nd | 7.80 | nd | nd | nd |
| eIF3b/eIF3a-cterm | nd | nd | 7.25 | nd | nd |
| eIF3b-cterm/eIF3i/eIF3g | nd | nd | 12.20 | nd | nd |

*nd – not determined

DOI: https://doi.org/10.7554/eLife.39273.014

head and body) (*Figure 1—figure supplement 5*), as in previous py48S P$_{IN}$/closed complexes, and in contrast to the open-latch and relaxed h28 conformations of a py48S P$_{OUT}$/open complex (*Figure 1—figure supplement 5*) (*Hussain et al., 2014*; *Llácer et al., 2015*). Density for the eIF5-NTD is observed on the 40S platform near the P site (*Figure 2A*), which is also the site for binding of eIF1 (*Llácer et al., 2015*; *Hussain et al., 2014*; *Lomakin and Steitz, 2013*; *Hashem et al., 2013*; *Rabl et al., 2011*; *Aylett et al., 2015*). Although eIF1 and the eIF5-NTD share structural similarity (*Conte et al., 2006*) (*Figure 2—figure supplement 1A–B*), the resolution of the map allowed us to unambiguously determine that the density belongs to eIF5-NTD and not eIF1 (see below for details of fitting eIF5-NTD into the density map). Clear density for the codon:anticodon interaction is observed at the P site; and the complete N-terminal tail (NTT) of eIF1A was resolved, stabilizing the codon:anticodon helix as seen in earlier closed py48S complexes containing eIF1 instead of eIF5-NTD (*Llácer et al., 2015*; *Hussain et al., 2014*).

In Map 1, the occupancies for eIF3, eIF2γ, eIF2β and the acceptor arm of the tRNA$_i$ are rather low. In order to observe the location of these ligands on the py48S-eIF5N, a larger data set and extensive 3D classification using different masks was employed to obtain multiple py48S-eIF5N maps (Maps A, B, C1 and C2) showing clear densities for eIF1A, eIF3, TC (including eIF2β), eIF5 and mRNA (*Figure 1—figure supplement 1*; see Materials and methods). Densities corresponding to the eIF5-CTD and eIF4 factors were not observed in any of these py48S-eIF5N maps, which presumably reflects the flexibility or dynamic nature of these domains/factors in the 48S PIC during the later steps of initiation.

**Table 3.** Contribution of non-crosslinked (157,868 particles) and crosslinked (113,838 particles) datasets to each map.

| Structures<br>Number of particles (% of the total) and resolution in Å | Map 1 | Map A | Map B | Map C1 | Map C2 |
|---|---|---|---|---|---|
| Non-crosslinked | 157,868 (58%) ; 3.00 | 23,219 (8.5%) ; 3.70 | 25,761 (9.5%) ; 3.60 | 27,012 (10%) ; 3.60 | 99,229 (36.5%) ; 3.10 |
| Crosslinked | - | 30,651 (11.5%) ; 4.30 | 28,938 (10.5%) ; 4.35 | 47,760 (17.5%) ; 4.30 | 37,874 (13.5%) ; 4.05 |
| Merged | 157,868 (58%) ; 3.00 | 53,870 (20%) ; 3.50 | 54,699 (20%) ; 3.50 | 74,772 (27.5%) ; 3.50 | 137,103 (50%) ; 3.10 |

DOI: https://doi.org/10.7554/eLife.39273.015

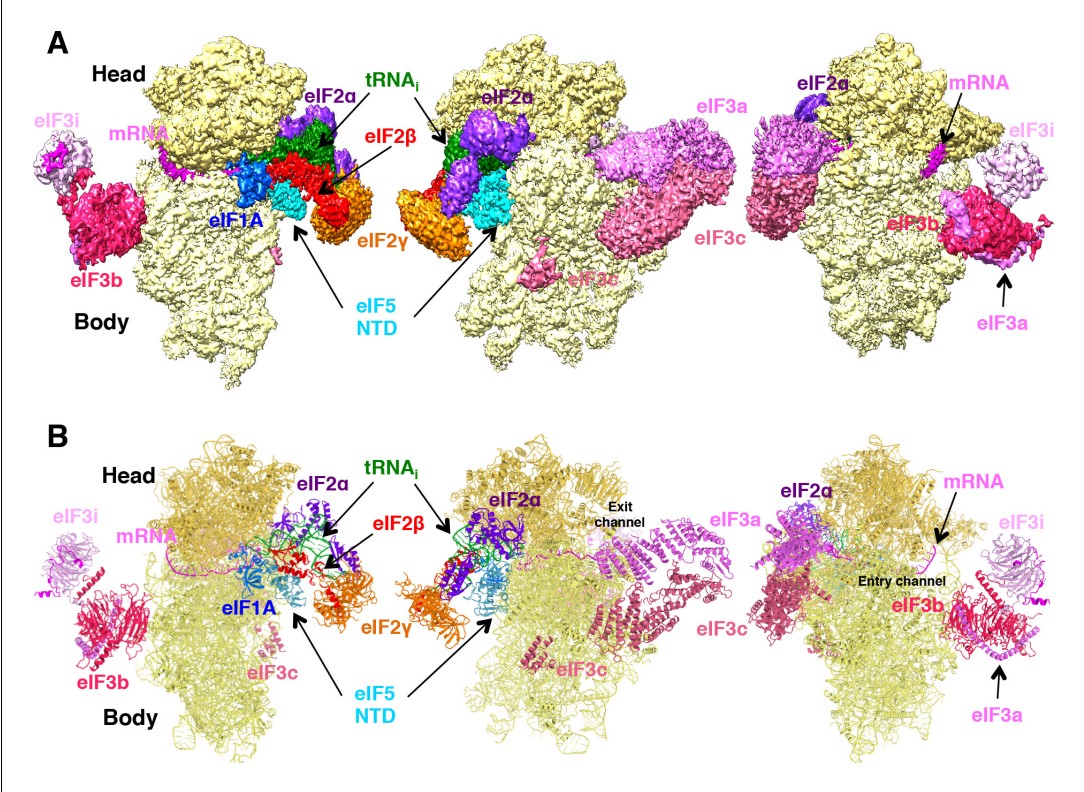

**Figure 1.** Cryo-EM structure of py48S-eIF5N. (**A**) CryoEM maps of the PIC py48S-eIF5N shown in three orientations. Regions of the map are colored by component to show the 40S subunit (yellow), eIF1A (blue), eIF5-NTD (cyan), Met-tRNAi$^{Met}$ (green), mRNA (magenta), eIF2α (violet), eIF2γ (orange), eIF2β (red), eIF3 (different shades of pink). The 40S head is shown in a darker yellow compared to the body. The density for 40S, eIF1A, mRNA, tRNA, eIF2 subunits and eIF5 is taken from Map C1, whereas density for eIF3 PCI domains is taken from Map A, and for eIF3-bgi subcomplex from Map B. The same colors are used in all the figures. (**B**) Atomic model for the PIC in the same colors and in the same three orientations. See also *Figure 1—figure supplements 1*, *2*, *3*, *4*, *5* and *6*.

DOI: https://doi.org/10.7554/eLife.39273.002

The following figure supplements are available for figure 1:

**Figure supplement 1.** Scheme of 3D classification of data.
DOI: https://doi.org/10.7554/eLife.39273.003

**Figure supplement 2.** Validation of the maps.
DOI: https://doi.org/10.7554/eLife.39273.004

**Figure supplement 3.** Fitting of ligands in density maps.
DOI: https://doi.org/10.7554/eLife.39273.005

**Figure supplement 4.** Map quality and local resolution Surface (left or top) and cross-sections (right or bottom) of gaussian-filtered maps, colored according to local resolution.
DOI: https://doi.org/10.7554/eLife.39273.006

**Figure supplement 5.** Latch and h28 conformation and head closure in different py48S PICs.
DOI: https://doi.org/10.7554/eLife.39273.007

**Figure supplement 6.** Comparison of the maps obtained with particles from sample 1 (non-crosslinked) and sample 2 (1%-formaldehyde crosslinked).
DOI: https://doi.org/10.7554/eLife.39273.008

## The eIF5-NTD replaces eIF1 on the 40S platform near the P site

A clear and distinct density for an 'eIF1-like' domain was observed at the top of h44 near the P site (*Figure 2A*), but fitting the known eIF1 structure into this density as seen in previous py48S-maps (*Hussain et al., 2014*; *Llácer et al., 2015*) could not account for all of it (*Figure 2B*). Also, there was no density to account for the C-terminal β-strand of eIF1. Moreover, close inspection revealed discrepancies between the densities and side chains of eIF1, particularly for β-hairpin one at the P site. Together, the known structural similarity of the eIF5-NTD with eIF1 (*Conte et al., 2006*) (*Figure 2—figure supplement 1A–B*), our previous suggestion that the eIF5-NTD might occupy the position of

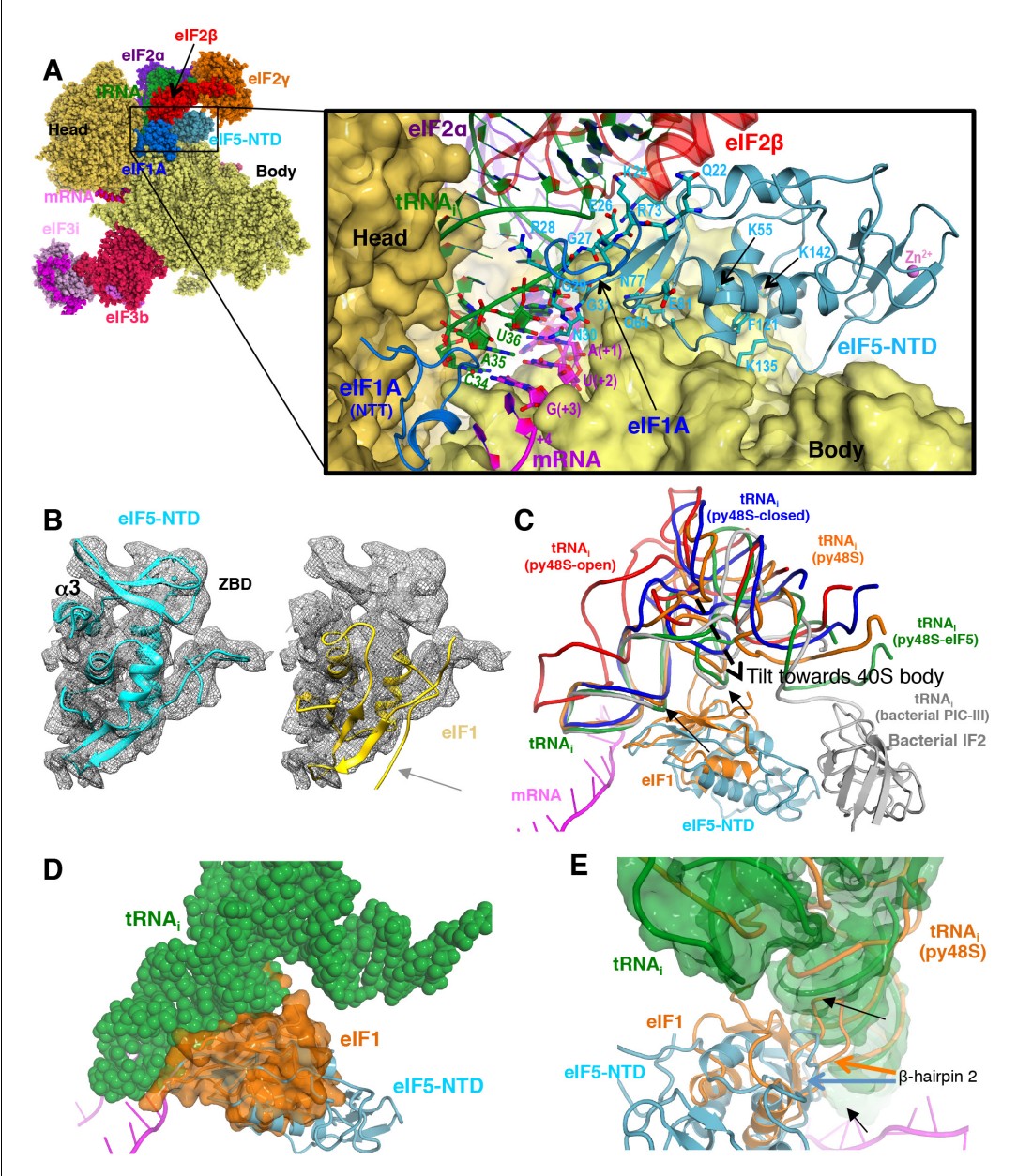

**Figure 2.** Contacts of eIF5-NTD with the other components in the 48S PIC. (**A**) A detailed view of the contacts of eIF5-NTD near the P site with the 40S subunit, tRNA$_i$, mRNA, eIF1A and eIF2β. eIF5 residues involved in the contacts are shown in sticks. (**B**) Fitting of eIF5-NTD (left) and eIF1(right) on density in Map 1(low-pass filtered to 5 Å). Zinc binding domain (ZBD) and helix α3 of eIF5, both absent in eIF1, are labeled. β5 of eIF1, which is not present in eIF5 is highlighted by a grey arrow. (**C**) The relative movement of the initiator tRNA in all reported yeast PICs, as deduced by a superposition using the 40S body. tRNA$_i$s from py48S-eIF5N (this study; green), py48S PIC (PDB 3J81; orange), py48S PIC-closed (PDB 3JAP; blue) and py48S PIC-open (PDB 3JAQ red) are shown. eIF5-NTD from py48S-eIF5N and eIF1 from py48S PIC (PDB 3J81) are also shown. For comparison, tRNA$_i$ and IF2 from a bacterial PIC with accommodated P site tRNA conformation is also shown (PICIII; PDB 5LMV; grey). In all closed conformations, the tip of the ASL is essentially in the same position; however, there is a different tilting of the tRNA$_i$ toward the 40S body in the different PICs. eIF1 would clash with tRNA$_i$ in py48S-eIF5N; black arrows highlight these clashes. (**D**) Representation of how eIF1 in py48S (transparent orange surface) would clash with tRNA$_i$ in py48S-eIF5N (in spheres). The model results from aligning the 40S bodies of the two structures. (**E**) eIF1 and eIF5-NTD share a similar fold; however, β-hairpin two in eIF5 is shorter than that in eIF1, which allows a further accommodation of tRNA$_i$ in the P site. eIF1 and tRNA$_i$ from py48S (PDB 3J81; orange) are superimposed on eIF5-NTD/tRNA$_i$ from py48S-eIF5N. See also *Figure 2—figure supplements 1*, *2* and *3*.

DOI: https://doi.org/10.7554/eLife.39273.009

The following figure supplements are available for figure 2:

**Figure supplement 1.** eIF5 and eIF1 comparison.

*Figure 2 continued on next page*

*Figure 2 continued*

DOI: https://doi.org/10.7554/eLife.39273.010

**Figure supplement 2.** Density of β-hairpins 1 and 2 of eIF5-NTD and its contacts with tRNA$_i$.

DOI: https://doi.org/10.7554/eLife.39273.011

**Figure supplement 3.** Comparison of eukaryotic and bacterial initiation following start-codon recognition.

DOI: https://doi.org/10.7554/eLife.39273.012

eIF1 on the 40S following eIF1 dissociation (*Nanda et al., 2009*), and our previous demonstration that the eIF5-NTD can bind directly to the 40S subunit (*Nanda et al., 2013*), all prompted us to place the structure of eIF5-NTD (PDB: 2E9H) into the unassigned density on the 40S platform. The eIF5-NTD structure accounted for the entire density, including the zinc-binding domain (ZBD) absent in eIF1 (*Figure 2B*), and the high resolution of the map enabled us to unambiguously model the eIF5-NTD at the atomic level (*Figure 1—figure supplement 3C* and *Figure 2—figure supplement 2*).

The eIF5-NTD binds on the platform at essentially the same location occupied by eIF1 in previous py48S structures (*Figure 2A,C* and *Figure 2—figure supplement 1C–F* and *Video 1*), interacting with 18S rRNA residues in h44 (1760; *S. cerevisiae* numbering), h45 (1780 and 1781) and h24 (994, 995, 1001, 1002 and 1004). In this position, eIF5-NTD interacts with eIF1A, as does eIF1 in other py48S structures; and also makes limited contacts with eIF2β and eIF2γ (*Figure 2—figure supplement 1C*). However, residue Arg15 of eIF5-NTD (essential for its GAP activity; (*Algire et al., 2005*)) is positioned more than 10 Å away from the bound GTP analog in eIF2γ (*Figure 2—figure supplement 1D*). Accordingly, this position and conformation of the eIF5-NTD does not appear compatible with the GAP activity of eIF5. Given that GTP hydrolysis occurs in the scanning complex but P$_i$ release requires eIF1 dissociation (*Algire et al., 2005*), and noting that eIF1 is absent and replaced by eIF5-NTD, we presume that this complex represents a state following both GTP hydrolysis and eIF1 dissociation but that the use of non-hydrolyzable GDPCP has prevented P$_i$ release.

Multiple residues in the eIF5-NTD, including Lys24, Gly27, Arg28, Gly29, Asn30, and Gly31 (in β-hairpin 1), and Lys71 and Arg73 (in β-hairpin 2), make multiple contacts with the anticodon stem loop (ASL) of the tRNA$_i$ (*Figure 2A*, *Figure 2—figure supplement 1C*, *Figure 2—figure supplement 2A,B*), and these contacts are more extensive and more favorable than are those made by the structurally analogous β-hairpins 1 and 2 of eIF1 in py48S (*Figure 2—figure supplement 1E,F* and *Video 1*) (*Hussain et al., 2014*). Interestingly, β-hairpin 1 of eIF5-NTD is positioned in the mRNA channel at the P site and monitors the codon:anticodon interaction in a similar fashion as the β-hairpin 1 of eIF1 (*Figure 2A,C*)(*Hussain et al., 2014*; *Martin-Marcos et al., 2013*). The conserved Asn30 in β-hairpin 1 of eIF5 makes contacts with both the codon and anticodon (*Figure 2A,C*, *Figure 2—figure supplement 2A*). However, in eIF5-NTD the β-hairpin one is shorter and contains three Gly residues compared to only one Gly in the case of eIF1. Gly27, Gly29 and Gly31 are closely packed against the ASL (*Figure 2A*) and any larger residue would create a steric clash with the ASL. The β-hairpin 2 of eIF5-NTD is also three residues shorter than that of eIF1 and is oriented away from the tRNA$_i$ to allow the latter to be tilted more toward the 40S body compared to previous py48S complexes that contain eIF1 (*Hussain et al., 2014*) (*Figure 2C–E* and *Video 1*).

A superimposition of this structure containing eIF5-NTD with the previous py48S structure containing eIF1 (*Hussain et al., 2014*) reveals that eIF1 would sterically clash with tRNA$_i$ at its position in py48S-eIF5N (*Figure 2D–E* and *Video 1*) indicating a further accommodation of the tRNA$_i$

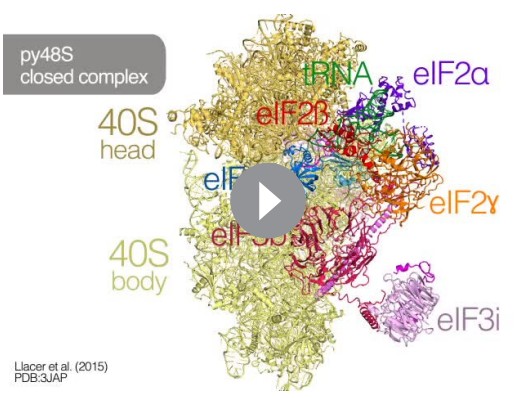

**Video 1.** Movie highlighting tRNA accommodation, leading to eIF1 dissociation and eIF5-NTD recruitment to the 48S complex. Detailed contacts of eIF5-NTD with other elements of the 48S complex are also shown.

DOI: https://doi.org/10.7554/eLife.39273.016

in the P site after dissociation of eIF1. Of the various eukaryotic py48S complexes containing eIF2, the structure here shows the maximum degree of tRNA$_i$ accommodation and tilt toward the 40S body (*Figure 2C*). This tRNA$_i$ tilt toward the body is also similar to that found in eukaryotic 80S initiation complexes containing eIF5B (*Figure 2—figure supplement 3A*) (*Fernández et al., 2013*; *Yamamoto et al., 2014*). Given that the affinity of TC for the PIC increases when eIF1 is ejected after AUG recognition (*Passmore et al., 2007*; *Nanda et al., 2013*), the tRNA$_i$ conformation observed in the present complex probably represents its most stable conformation, and it is conceivable that eIF5-NTD participates in this stabilization via its interaction with the ASL.

## eIF5-NTD substitutions at the codon:anticodon interface alter the stringency of AUG start-codon selection *in vivo*

To examine the physiological significance of the direct contacts observed between the eIF5-NTD and the start codon and ASL, specific residues were selected for mutagenesis based on their proximity to tRNA$_i$ or 18S rRNA in the PIC (*Figure 3A–B* and *Figure 3—figure supplement 1*). Mutations were generated in the gene coding for eIF5 harboring a C-terminal FLAG epitope (the *TIF5-FL* allele), on a *LEU2* plasmid, and introduced into a *his4-301 tif5Δ* strain lacking chromosomal *TIF5* and carrying WT *TIF5* on a *URA3* vector. The *his4-301* mutation confers auxotrophy for histidine (His$^-$) owing to the absence of the AUG start codon of the WT *HIS4* allele, which can be suppressed by error-inducing Sui$^-$ mutations that allow utilization of the third, UUG triplet as start codon, including the eIF1 mutation *sui1-L96P* (*Martin-Marcos et al., 2011*) (*Figure 3—figure supplement 2A*). Hence, by determining effects of the eIF5 mutations on the histidine requirement of the resulting *his4-301* strains, we could determine their effects on accurate start-codon recognition *in vivo* (*Figure 3—figure supplement 1*).

After plasmid-shuffling to evict WT *TIF5,* we found that all mutant strains were viable, but that the strains carrying *TIF5-FL* alleles *-R28E, -N30E, -K55E,* and *-K142E* displayed slow-growth (Slg$^-$) phenotypes of varying degrees compared to the WT *TIF5-FL* strain (*Figure 3C*;+His (D1)). None of the mutations conferred any marked differences in steady-state expression of the FLAG-tagged eIF5 proteins (*Figure 3—figure supplement 2B,C*). None of the mutants exhibited a His$^+$ phenotype on media lacking His or containing only 1% of the His used to fully supplement the auxotrophy (data not shown), suggesting the absence of marked Sui$^-$ hypoaccuracy phenotypes. However, assaying β-galactosidase expressed from matched *HIS4-lacZ* reporters with either AUG or UUG start codons revealed that *E26K, K142E, G29R* and *N30R* conferred ~2-, 3-, 4- and 8-fold increases in the UUG: AUG ratio, respectively (*Figure 3C*, *HIS4-lacZ* UUG:AUG, rows 7–10 vs. 1). The absence of a His$^+$ phenotype for these mutations might result from a failure to increase the UUG:AUG ratio above a critical threshold level (*Martin-Marcos et al., 2013*); the strong slow-growth (Slg$^-$) phenotype of *K142E* would likely also impede growth on –His or 1% His medium.

Interestingly, the structure reveals that residues E26, G29 and N30 are in proximity to the ASL of tRNA$_i$ (*Figure 3A–B*), and we hypothesized that increasing the positive charge of these residues by replacing them with R or K might stabilize the P$_{IN}$ state of TC binding even on near-cognate UUG start codons, thus accounting for the increased UUG:AUG initiation ratios conferred by the E26K, G29R, and N30R substitutions (*Figure 3C*). If so, then decreasing the positive charge of residues 28 and 73, which also approach the tRNA$_i$ ASL, by the *R28A, R28E,* and *R73A* mutations, might destabilize the P$_{IN}$ state and increase discrimination against UUG start codons. To test this idea, the *TIF5-FL* strains were transformed with plasmid-borne *SUI3-2*, encoding the S264Y substitution in eIF2β (eIF2β-S264Y) that confers a dominant His$^+$/Sui$^-$ hypoaccuracy phenotype and elevates the UUG: AUG ratio in otherwise WT strains (*Huang et al., 1997*).

As expected, *SUI3-2* confers growth on media containing 1% His and elevates the UUG:AUG ratio by ~5 fold (*Figure 3D*, *HIS4-lacZ* UUG:AUG rows 1–2). Importantly, the His$^+$/Sui$^-$ hypoaccuracy phenotype of *SUI3-2* is suppressed efficiently by the *-R28E, -R28A,* and *-R73A* alleles of *TIF5-FL*, which also substantially diminish the UUG:AUG ratio in *SUI3-2* cells (*Figure 3D*, rows 3 and 7 vs. 2; *Figure 3—figure supplement 2E*, rows 2 and 4), thus conferring Ssu$^-$ hyperaccuracy phenotypes. Interestingly, *G29E, N30E, K55E* and *K142E* also suppress the His$^+$/Sui$^-$ hypoaccuracy phenotype and mitigate the elevated UUG:AUG ratio conferred by *SUI3-2* (*Figure 3D*, rows 4–6, 8 vs. 2). The Ssu$^-$ hyperaccuracy phenotypes of *R28E, N30E* and *K55E* are dominant in strains harboring *SUI3-2* and WT *TIF5* (*Figure 3—figure supplement 2D*), suggesting that these eIF5 variants can efficiently

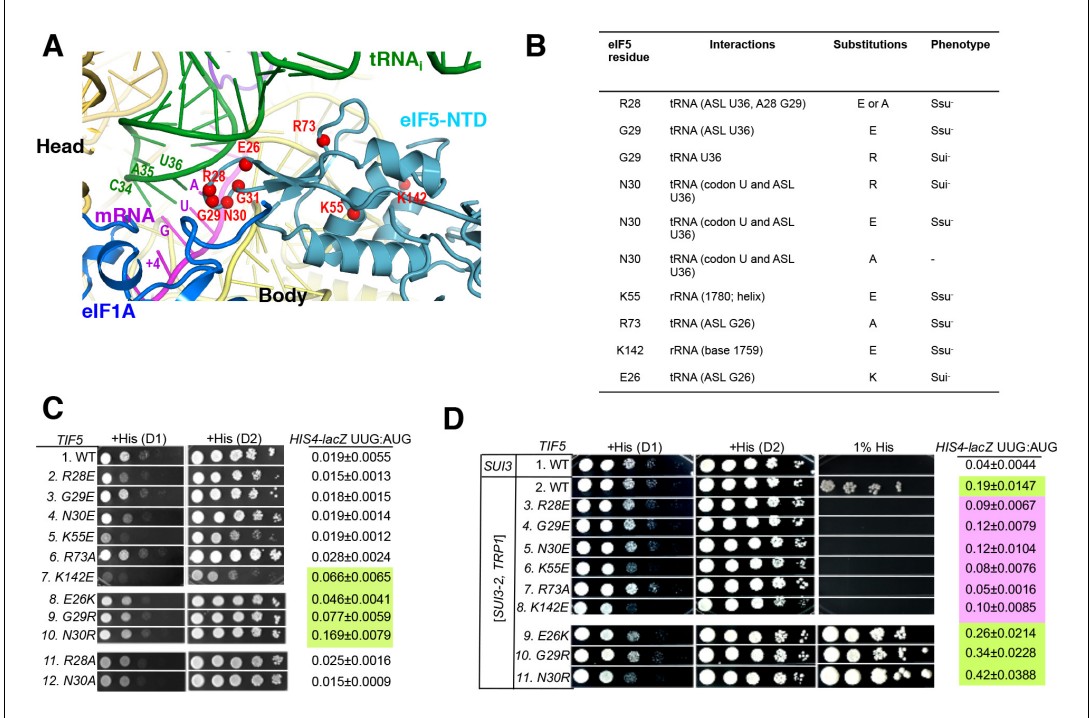

**Figure 3.** Genetic evidence that contacts of the eIF5-NTD with the tRNA_i ASL and mRNA AUG codon in py48S-eIF5N are crucial for stringency of start-codon recognition *in vivo*. (**A**) Location of the eIF5 residues substituted in genetic studies, highlighted in red and shown as spheres. (**B**) Summary of eIF5-NTD residues substituted by *TIF5* mutations (col. 1), their interactions in py48S-eIF5N (col. 2), the amino acid substitutions introduced (col. 3), and the observed Sui⁻ or Ssu⁻ phenotypes *in vivo* (col. 4) revealed by results in (C or D). (**C**) Slg⁻ and His⁺/Sui⁻ phenotypes were determined for derivatives of *his4-301 tif5Δ* strain ASY100 harboring the indicated *TIF5-FL* alleles on *LEU2* plasmids. Ten-fold serial dilutions of the strains were spotted on synthetic complete medium lacking leucine (SC-L) and supplemented with 0.3 mM histidine (+His), and incubated for 1d (D1) or 2d (D2) at 30°C. To assess the effect of *TIF5* substitutions on start-codon recognition, the strains were transformed with *HIS4-lacZ* reporter plasmids containing AUG (p367) or UUG (p391) start codons. Cells were cultured in SC medium lacking leucine and uracil at 30°C and β-galactosidase activities were measured in whole cell extracts (WCEs). Ratios of β-galactosidase expressed from the UUG to AUG reporter were calculated from four independent transformants and mean ratios and S.E.M.s (error bars) are reported on the right. Ratios indicating Sui⁻ phenotypes are highlighted in lime green. (**D**) Strains described in (**C**) were transformed with *SUI3-2* plasmid pRSSUI3-S264Y-W (rows 2–11) or empty vector (row 1). Slg⁻ and His⁺/Sui⁻ phenotypes were determined by spotting the 10-fold serial dilutions of strains on SC medium lacking leucine and tryptophan and supplemented with either 0.3 mM histidine (+His) or 0.0003 mM histidine (1% His), and incubated for 1d (D1) or 2d (D2) for +His medium and 6d for 1% His medium, at 30°C. The *HIS4-lacZ* initiation ratios, were determined as in (**B**) except the cells were grown in SC medium lacking leucine, uracil and tryptophan. Ratios indicating Sui⁻ (lime green) or Ssu⁻ (pink) phenotypes are highlighted. See also *Figure 3—figure supplements 1* and *2*.

DOI: https://doi.org/10.7554/eLife.39273.017

The following figure supplements are available for figure 3:

**Figure supplement 1.** Genetic assays for mutations altering the accuracy of start-codon selection *in vivo*.
DOI: https://doi.org/10.7554/eLife.39273.018

**Figure supplement 2.** Genetic assays and eIF5 variants expression.
DOI: https://doi.org/10.7554/eLife.39273.019

compete with WT eIF5 for incorporation into PICs but function poorly in stabilizing the P_IN state at UUG codons.

The *TIF5-FL* alleles *E26K*, *G29R* and *N30R*, which elevate the UUG:AUG ratio in otherwise WT cells (*Figure 3C*, rows 7–10 vs. 1), also exacerbate the His⁺/Sui⁻ hypoaccuracy defect of *SUI3-2* (*Figure 3D*, 1% His, rows 9–11 vs. 2) and confer ~1.4-, 1.8- and 2.2-fold increases in the UUG:AUG ratio compared to *SUI3-2* cells containing WT *TIF5-FL* (*Figure 3D*, *HIS4-lacZ* UUG:AUG rows 9–11 vs. 2). TIF5-FL-N30R also conferred a modest slow growth phenotype in *SUI3-2* cells (*Figure 3D*, +His, row 11 vs. 2), which was not seen in otherwise WT cells containing this allele (*Figure 3C*, row 10 vs. 1). The ability of *N30R* and *E26K* to intensify the His⁺/Sui⁻ hypoaccuracy defect of *SUI3-2* is dominant, occurring in strains harboring WT *TIF5* (*Figure 3—figure supplement 2C*, rows 6–7 vs. 2),

indicating that these eIF5 variants can also compete with WT eIF5 for incorporation into the PIC and stabilize the $P_{IN}$ state at UUG codons.

It is noteworthy that substitutions of G29 introducing positively charged residues decrease initiation accuracy, whereas substitutions with negatively charged residues increase accuracy (*Figure 3C–D*). These findings are consistent with the idea that introducing a positive charge at the interface with tRNA$_i$ enhances electrostatic attraction with the ASL to stabilize the $P_{IN}$ state at UUG start codons, whereas a negatively charged side-chain at this position destabilizes $P_{IN}$ through electrostatic repulsion with the ASL to preferentially diminish selection of UUG codons, which form mismatched duplexes with the tRNA$_i$ ASL. The same reasoning can explain the opposite phenotypes of the Arg and Glu substitutions of N30 (*Figure 3C–D*, *N30E, N30R*); and as noted above, the hyperaccuracy phenotype of R28E and hypoaccuracy phenotype of E26K. The fact that R73A and R28A also confer hyperaccuary phenotypes without introducing electrostatic repulsion underscores the importance of the native contacts of R28 and R73 with the tRNA$_i$ ASL in stabilizing $P_{IN}$. Together, the genetic data provide strong evidence that the contacts between the eIF5-NTD and the tRNA$_i$ ASL visualized in the cryo-EM structure are crucial for a WT stringency of start-codon recognition *in vivo*.

## eIF5-NTD substitutions at the codon:anticodon interface alter the influence of the start codon on transition to the closed PIC conformation

We have previously shown that monitoring dissociation of fluorescently labeled eIF1A from 48S PICs using fluorescence anisotropy is a useful tool to distinguish between open/$P_{OUT}$ and closed/$P_{IN}$ conformations of the PIC (*Saini et al., 2014*; *Maag et al., 2006*). Although dissociation of eIF1A from the PIC at this stage of the initiation process is slow and does not appear to be a physiologically relevant event, it does report on the relative abundance and stability of the open and closed states of the complex (*Figure 4A*). In WT PICs, dissociation of eIF1A occurs with biphasic kinetics, with the fast phase reflecting complexes in the open state, in which eIF1A is less stably bound, and the slow phase reflecting the more stable, closed state. The ratio of amplitudes of the slower phase ($a_2$) over the fast phase ($a_1$) is taken as the apparent equilibrium constant between the closed and open states ($a_2/a_1$) and is referred to as '$K_{amp}$'. As observed in previous studies, with WT complexes $K_{amp}$ is higher when the model mRNA has an AUG start codon (mRNA(AUG)) than when it has a near-cognate UUG codon (mRNA(UUG)) (5.9 vs. 3.2; *Table 4A*, rows 1–2), consistent with the closed state being more favored in the former case than in the latter. This effect is also reflected in the rate constants for the fast ($k_1$) and slow ($k_2$) phases, which are both higher for complexes assembled on mRNA(UUG) than on mRNA(AUG) (6 vs. 22 $\times$ 10$^{-3}$ s$^{-1}$ and 0.4 vs. 2.1 $\times$ 10$^{-3}$ s$^{-1}$, respectively; *Table 4A* and *Figure 4B*), indicating that eIF1A is less stably bound in both the open and closed states in complexes assembled on a near-cognate start codon. Consistent with this interpretation, the fluorescence anisotropy of the C-terminal fluorescein moiety on eIF1A is higher in complexes assembled on mRNA(AUG) ($R_{bound} = 0.21$) than on mRNA(UUG) ($R_{bound} = 0.18$) (*Table 4A*, rows 1–2). Because higher fluorescence anisotropies indicate less freedom of rotation of the fluorophore, these data indicate that a 'tighter', more constrained complex is preferentially formed on mRNA containing the cognate AUG codon.

Using this system, we sought to determine the mechanistic impact of substitutions in the eIF5 residues that are in proximity to the start codon:tRNA$_i$ anticodon helix. As described above, in the presence of WT eIF5, eIF1A is more stably bound to the PIC with mRNA(AUG) than mRNA(UUG) (*Figure 4B* blue closed circles versus blue closed squares) with a relatively higher $K_{amp}$ for mRNA(AUG) (*Table 4A*; rows 1–2), indicating a greater preponderance of complexes in the closed state. As observed previously (*Saini et al., 2014*; *Maag et al., 2006*), the G31R-eIF5 substitution, which has a strong, dominant Sui$^-$ hypoaccuracy phenotype *in vivo*, inverts the effect of an AUG versus UUG start codon on the dissociation kinetics (*Figure 4B* red closed circles *versus* red closed squares). In contrast to WT eIF5, PICs containing G31R eIF5 have higher $K_{amp}$ values at UUG than AUG start codons (*Table 4A*, rows 3–4), thus indicating the closed state of the PIC is favored in the former case relative to the latter, consistent with the Sui$^-$ hypoaccuracy phenotype. Similarly, $k_1$ and $k_2$ values are lower for PICs assembled with G31R on UUG start codons than on AUG codons ($k_1$ values 18 and 7 $\times$ 10$^{-3}$ s$^{-1}$, and $k_2$ values 3.0 and 0.5 $\times$ 10$^{-3}$ s$^{-1}$, for AUG and UUG, respectively; *Table 4A*, rows 3–4 *versus* 1–2). The $R_{bound}$ values also invert, becoming 0.19 and 0.20 for AUG and UUG, respectively (*Table 4A*, rows 3–4 *versus* 1–2). These results are consistent with the placement

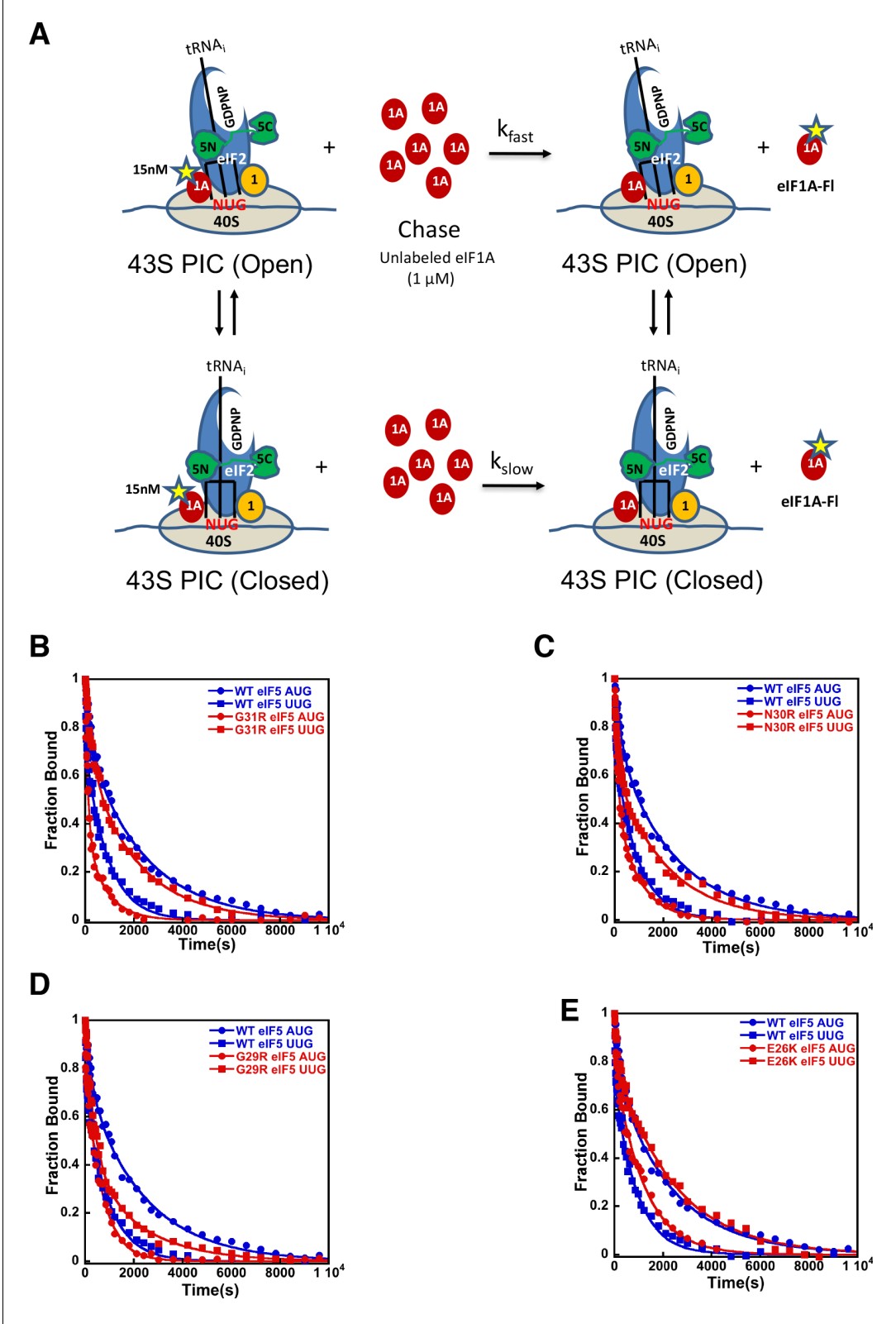

**Figure 4.** eIF5-NTD Sui⁻ mutants stabilize the closed conformation of the PIC at UUG codons. Dissociation of fluorescein-tagged eIF1A from 43S·mRNA complexes reconstituted with model mRNAs containing an AUG or UUG start codon and either WT or Sui⁻ variants of eIF5 was monitored as decrease in fluorescence anisotropy over time after addition of excess unlabeled eIF1A. The data were fit with a double exponential decay equation. (**A**)

*Figure 4 continued on next page*

*Figure 4 continued*

Schematic of the eIF1A dissociation assay. PICs are assembled with fluorescein-labeled eIF1A (15 nM) and then dissociation of the labeled factor is initiatied with a chase of 66-fold excess unlabeled eIF1A (1 μM). The fast phase ($k_{fast}$) of the dissociation of eIF1A-Fl reflects release from the open state of the PIC, where as the second, slower phase ($k_{slow}$) reflects dissociation from the closed state of the PIC. (B) eIF1A dissociation from 48S PICs assembled with WT eIF5 (blue) or eIF5-G31R (red) and mRNAs with an AUG (closed circles) or UUG (closed squares) start codons. (C–E) eIF1A dissociation from PICs containing WT eIF5 (blue) or the indicated mutant eIF5 (red) and mRNAs with AUG (closed circles) or UUG (closed squares) start codons, for eIF5-N30R (C), eIF5-G29R (D), or eIF5-E26K (E). The curves shown are representative experiments. Mean kinetic parameters and average deviations from multiple, independent experiments are presented in *Table 4*.

DOI: https://doi.org/10.7554/eLife.39273.020

The following source data is available for figure 4:

**Source data 1.** Kinetic parameters for dissociation of eIF1A from 48S PIC.

DOI: https://doi.org/10.7554/eLife.39273.021

of the G31 residue directly across from the first position of the codon:anticodon helix (*Figure 2A*), where an arginine substitution could stabilize the formation of a U:U mismatch and the closed/$P_{IN}$ state of the complex.

**Table 4.** Kinetic parameters for dissociation of eIF1A from 48S PIC

**(A) Kinetic parameters for dissociation of eIF1A from 48S PIC in presence of eIF5 Sui⁻ mutants**

| eIF5 variants | mRNA | $k_1$ (open) ($\times 10^{-3}$ s$^{-1}$) | $k_2$ (closed) ($\times 10^{-3}$ s$^{-1}$) | $a_1$ (open) | $a_2$ (closed) | $K_{amp}$* ($a_2/a_1$) | $R_{bound}$[†] |
|---|---|---|---|---|---|---|---|
| WT | AUG | 6 ± 1 | 0.4 ± 0.05 | 0.15 ± 0.02 | 0.85 ± 0.02 | 5.9 | 0.2105 ± 0.002 |
| | UUG | 22 ± 4 | 2.1 ± 0.3 | 0.24 ± 0.02 | 0.76 ± 0.02 | 3.2 | 0.1820 ± 0.002 |
| G31R | AUG | 18 ± 3 | 3.0 ± 0.4 | 0.33 ± 0.03 | 0.67 ± 0.03 | 2.1 | 0.1915 ± 0.001 |
| | UUG | 7.0 ± 1.5 | 0.5 ± 0.02 | 0.13 ± 0.01 | 0.87 ± 0.02 | 6.9 | 0.2025 ± 0.001 |
| N30R | AUG | 10.0 ± 1.0 | 1.5 ± 0.3 | 0.50 ± 0.1 | 0.50 ± 0.1 | 1.0 | 0.2030 ± 0.002 |
| | UUG | 6.0 ± 1.0 | 0.5 ± 0.1 | 0.30 ± 0.05 | 0.70 ± 0.05 | 2.3 | 0.2100 ± 0.002 |
| G29R | AUG | 20 ± 3.0 | 1.5 ± 0.4 | 0.40 ± 0.02 | 0.60 ± 0.02 | 1.5 | 0.1885 ± 0.001 |
| | UUG | 6.0 ± 1.6 | 0.6 ± 0.1 | 0.20 ± 0.03 | 0.80 ± 0.03 | 4.0 | 0.1925 ± 0.002 |
| E26K | AUG | 17 ± 2.6 | 0.9 ± 0.05 | 0.23 ± 0.02 | 0.77 ± 0.02 | 3.5 | 0.1835 ± 0.001 |
| | UUG | 8.0 ± 1.5 | 0.4 ± 0.05 | 0.16 ± 0.01 | 0.84 ± 0.01 | 5.2 | 0.1900 ± 0.002 |

**(B) Kinetic parameters for dissociation of eIF1A from 48S PIC in presence of eIF5 Ssu⁻ and Sui3-2 eIF2**

| eIF5 variants; Sui 3–2 eIF2 | mRNA | $k_1$(open) ($\times 10^{-3}$ s$^{-1}$) | $k_2$(closed) ($\times 10^{-3}$ s$^{-1}$) | $a_1$(open) | $a_2$(closed) | $K_{amp}$* ($a_2/a_1$) | $R_{bound}$[†] |
|---|---|---|---|---|---|---|---|
| Sui3-2 | AUG | 4 ± 1 | 0.3 ± 0.04 | 0.22 ± 0.03 | 0.78 ± 0.03 | 3.4 | 0.1945 ± 0.002 |
| | UUG | 5 ± 1.8 | 0.5 ± 0.04 | 0.20 ± 0.02 | 0.80 ± 0.02 | 4.0 | 0.1910 ± 0.001 |
| N30E | AUG | 6 ± 0.5 | 0.4 ± 0.04 | 0.15 ± 0.02 | 0.85 ± 0.01 | 5.6 | 0.2045 ± 0.0005 |
| | UUG | 15 ± 4 | 1.0 ± 0.16 | 0.23 ± 0.01 | 0.77 ± 0.01 | 3.3 | 0.1830 ± 0.003 |
| G29E | AUG | 8.0 ± 1.0 | 0.4 ± 0.02 | 0.22 ± 0.02 | 0.78 ± 0.02 | 3.5 | 0.2084 ± 0.0006 |
| | UUG | 18 ± 3.0 | 0.36 ± 0.03 | 0.28 ± 0.02 | 0.72 ± 0.02 | 2.6 | 0.1830 ± 0.003 |
| R28E | AUG | 13 ± 4.0 | 0.5 ± 0.04 | 0.21 ± 0.01 | 0.80 ± 0.01 | 4.0 | 0.2040 ± 0.001 |
| | UUG | 23 ± 2.0 | 0.9 ± 0.06 | 0.25 ± 0.03 | 0.75 ± 0.03 | 3.0 | 0.1910 ± 0.003 |
| R28A | AUG | 8.0 ± 0.8 | 0.3 ± 0.04 | 0.20 ± 0.02 | 0.80 ± 0.02 | 4.0 | 0.2105 ± 0.001 |
| | UUG | 14 ± 4.0 | 0.7 ± 0.08 | 0.24 ± 0.01 | 0.76 ± 0.01 | 3.2 | 0.1810 ± 0.003 |

*Higher values of $K_{amp}$ indicate that a greater proportion of the complexes are in the closed state (*Saini et al., 2014*).

†Higher values of $R_{bound}$ indicate that more complexes are in the constrained, closed state (*Saini et al., 2014*).

DOI: https://doi.org/10.7554/eLife.39273.022

The additional Sui⁻ hypoaccuracy substitutions in the eIF5-NTD generated here, E26K, G29R, and N30R, all produced a similar pattern to what we observed for eIF5-G31R (*Figure 4B–E*). All three substitutions led to higher $K_{amp}$ values in PICs with mRNA(UUG) versus mRNA(AUG) (*Table 4A*, rows 5–10), implying that these substitutions in eIF5, like G31R, shift the equilibrium more toward the closed/$P_{IN}$ state at the near-cognate UUG start codon and away from it at AUG codons. As with eIF5-G31R, $k_1$ and $k_2$ were both lower on UUG codons than AUG codons for all three mutants (*Table 4A*). $R_{bound}$ values were either equal for UUG and AUG codons (G29R) or greater for UUG codons (N30R and E26K), as was also the case for G31R (*Table 4A*). These results are consistent with the proposal that these positive charge substitutions in eIF5-NTD in the vicinity of the codon:anticodon helix electrostatically stabilize the $P_{IN}$ conformation on near-cognate codons. This does not seem to be the only effect, however, because these substitutions actually appear to destabilize the closed/$P_{IN}$ state on AUG codons, possibly due to steric clashes with the A:U base pair introduced by the arginine side chain.

We also determined the effects of substitutions in the eIF5-NTD at the same or nearby residues designed to decrease the positive charge or increase the negative charge: G29E, N30E, R28E and R28A. Because these substitutions produced hyperaccurate Ssu⁻ phenotypes in the genetic experiments described above, we examined their effects on eIF1A dissociation in the context of PICs assembled with *SUI3-2* (eIF2β-S264Y). Consistent with its Sui⁻ hypoaccuracy phenotype, we observed that eIF2β-S264Y slows the rate of eIF1A dissociation from 48S PICs assembled on mRNA(UUG), reducing the differences in rate constants, $K_{amp}$ and $R_{bound}$ values between 48S PICs on AUG and UUG mRNAs, relative to the WT complexes (*Figure 5A* red circles *versus* red squares; row 2 *Table 4A versus* row 2 *Table 4B*). These results suggest that the eIF2β-S264Y mutant of eIF2 stabilizes the closed/$P_{IN}$ state of the PIC at UUG codons.

With 48S PICs assembled with eIF2β-S264Y, the eIF5 variants R28A, G29E, and N30E increased the overall rate of eIF1A dissociation with mRNA(UUG) as compared to the native eIF5 (*Figure 5B–D*, blue squares versus red squares). These substitutions decrease the occupancy of the closed complex at UUG start codons, as indicated by the decreased $K_{amp}$ values on UUG relative to AUG start codons compared to the case with eIF2β-S264Y PICs containing WT eIF5 (*Table 4B*, rows 5–10). The N30E and R28E derivatives of eIF5 also increase $k_2$ values for complexes with UUG by ~2 fold (*Table 4B*, rows 3–4 and rows 7–10), indicating that these substitutions destabilize eIF1A binding to the closed state of the PIC at UUG start codons. All four hyperaccurate Ssu⁻ eIF5 variants also increase $k_1$ with both UUG and AUG codons, suggesting they destabilize eIF1A binding to the open state of the PIC. In all cases, the substitutions partially or completely restore the difference in $R_{bound}$ values between AUG and UUG complexes that was eliminated by the eIF2β-S264Y mutant with WT eIF5 (*Table 4B*, rows 3–10 versus 1–2). Taken together, these results suggest that the hyperaccurate Ssu⁻ eIF5 suppressors of *SUI3-2* eIF2 revert the equilibrium back toward the open/$P_{OUT}$ conformation of the PIC at UUG codons while promoting the closed/$P_{IN}$ conformation at AUG codons.

## eIF5-NTD substitutions at the codon:anticodon interface alter the coupling of $P_i$ release to start-codon recognition

We next checked the effect of the eIF5 substitutions on the rate of phosphate ($P_i$) release from eIF2 in the PIC in response to recognition of cognate AUG and near-cognate UUG start codons (*Figure 5E*). $P_i$ release is a late step in start-codon recognition and is gated by eIF1 release and movement of the eIF1A-CTT closer to the eIF5-NTD. It is thought to help commit the PIC to initiation at the selected point on the mRNA. Previous studies have shown that $P_i$ release is influenced by the nature of the start codon in the mRNA, with a higher rate observed from PICs assembled on AUG start codons as compared to PICs on near-cognate UUG codons (*Algire et al., 2005*; *Saini et al., 2014*).

In accordance with earlier studies (*Saini et al., 2014*; *Algire et al., 2005*), we observed that the kinetics of $P_i$ release is 2- to 3-fold faster in response to AUG as compared to UUG start codons ($k_{obs}$ values of 0.60 s$^{-1}$ *versus* 0.26 s$^{-1}$) (*Figure 5F*; *Table 5A*, row 1). This trend is reversed when the Sui⁻ hypoaccurate G31R eIF5 mutant replaces the WT factor, with a $k_{obs}$ of 0.6 s$^{-1}$ for PICs assembled on UUG codons versus 0.3 s$^{-1}$ for complexes on AUG codons (*Figure 5F*; *Table 5A*, row 2). This result is consistent with previous observations (*Saini et al., 2014*) and the Sui⁻ hypoaccuracy phenotype of the G31R mutant. Similarly, all of the new Sui⁻ hypoaccuracy substitutions in eIF5 (N30R, G29R, and E26K) suppress the rate of $P_i$ release from the PIC in response to recognition of

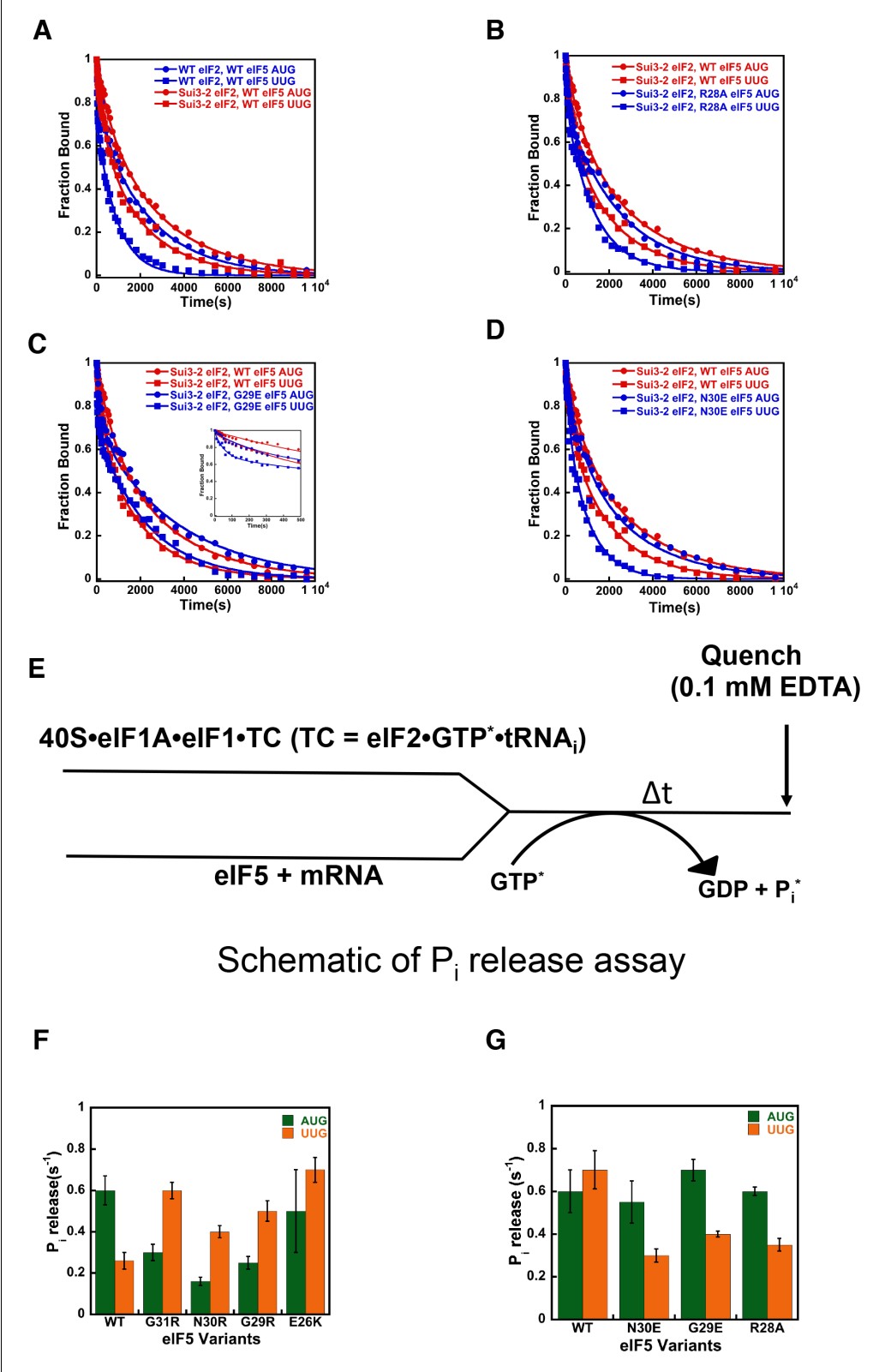

**Figure 5.** eIF5-NTD Ssu⁻ mutants destabilize the closed conformation of the PIC and accelerate $P_i$ release at UUG codons in the presence of the *SUI3-2* Sui⁻ variant of eIF2. (A–D) eIF1A dissociation kinetics experiments conducted as in *Figure 4* for PICs assembled with mRNAs containing AUG (closed circles) or UUG (closed squares) start codons and the following forms of eIF2 and eIF5: (A) WT eIF2/WT eIF5 (blue) or Sui3-2 eIF2/WT eIF5 (red); (B) Sui3-2 eIF2/WT eIF5 (red) or Sui3-2 eIF2/eIF5-R28A (blue); (C) Sui3-2 eIF2/WT eIF5 (red) or Sui3-2 eIF2/eIF5-G29E (blue); (D) Sui3-2 eIF2/WT eIF5 (red) or

*Figure 5 continued on next page*

*Figure 5 continued*

Sui3-2 eIF2/eIF5-N30E (blue). The curves shown are representative experiments. Mean kinetic parameters and average deviations from multiple, independent experiments are presented in *Table 4*. (E) Schematic of $P_i$ release experiment (see Materials and methods). (F–G) Rates of $P_i$ release from PICs assembled with mRNAs containing AUG (green) or UUG (orange) start codons and the indicated WT or mutant variants of eIF5. WT eIF2 was employed in (F), whereas Sui3-2 eIF2 was used in (G). Error bars depict average deviations.
DOI: https://doi.org/10.7554/eLife.39273.023

The following source data is available for figure 5:

**Source data 1.** eIF1A dissociation kinetics assays.
DOI: https://doi.org/10.7554/eLife.39273.024

cognate AUG start codons and/or enhance it in response to near-cognate UUG codons (*Figure 5F*, compare green and orange bars; *Table 5A*, rows 3–5). These results are consistent with the conclusion that these substitutions that increase the positive charge on the eIF5-NTD in the region of the codon:anticodon helix stabilize the closed state of the PIC at UUG codons, while destabilizing it at AUG codons. Thus, in accordance with the eIF1A dissociation kinetics results described above, the $P_i$ release kinetics for the positive charge eIF5 substitutions help explain their Sui⁻ hypoaccuracy phenotypes observed *in vivo*.

Next, we monitored the kinetics of AUG- and UUG-triggered $P_i$ release from PICs assembled with the Sui⁻ hypoaccuracy variant eIF2β-S264Y (encoded by *SUI3-2*) mutant and either WT eIF5 or one of the hyperaccurate Ssu⁻ variants described above (N30E, G29E, R28A). In agreement with the eIF1A dissociation kinetics results described above, the eIF2β-S264Y mutant normalizes the rate of $P_i$ release from complexes assembled on UUG and AUG start codons ($k_{obs}$ values of 0.7 and 0.6 s⁻¹, respectively; *Figure 5G*; *Table 5B*, row one *versus Table 5A*, row 1). Unlike the behavior of G31R eIF5, the eIF2β-S264Y mutant does not decrease the rate of $P_i$ release with AUG start codons but instead only increases the rate with UUG codons (*Figure 5G versus Figure 5F*), suggesting that it specifically enhances the stability of the closed/$P_{IN}$ conformation of the PIC at near-cognate (UUG) codons. Consistent with their effects in the eIF1A dissociation assay, the hyperaccurate Ssu⁻ eIF5 variants N30E, G29E and R28A all suppress the effect of eIF2β-S264Y by decreasing the rate of $P_i$ release from UUG start codons ~ 2 fold ($k_{obs}$ values between 0.3–0.4 s⁻¹, *Table 5B*, rows 2–4) versus the rate observed with WT eIF5 ($k_{obs}$ of 0.7 s⁻¹, *Table 5B*, row 1), restoring the preference for AUG start codons. These results support the proposal that these substitutions, which increase the negative charge or decrease the positive charge in this region of the eIF5-NTD, destabilize the closed/$P_{IN}$ state of the PIC at near-cognate codons.

**Table 5.** Kinetic parameters for Pi release from 48S PIC.

**(A) Kinetic parameters for $P_i$ release from 48S PIC with eIF5 Sui⁻ mutants.**

| eIF5 Variants | Rate of $P_i$ release (s⁻¹) AUG | UUG |
|---|---|---|
| WT | 0.60 ± 0.08 | 0.26 ± 0.04 |
| G31R | 0.30 ± 0.04 | 0.60 ± 0.04 |
| N30R | 0.16 ± 0.02 | 0.41 ± 0.03 |
| G29R | 0.25 ± 0.03 | 0.55 ± 0.05 |
| E26K | 0.50 ± 0.20 | 0.72 ± 0.06 |

**(B) Kinetic parameters for Pi release from 48S PIC with eIF5 Ssu- mutants in presence of Sui3-2 eIF2**

| eIF5 Variants; Sui3-2 eIF2 | Rate of $P_i$ release (s⁻¹) AUG | UUG |
|---|---|---|
| WT | 0.60 ± 0.10 | 0.75 ± 0.09 |
| N30E | 0.55 ± 0.15 | 0.30 ± 0.03 |
| G29E | 0.72 ± 0.04 | 0.40 ± 0.01 |
| R28A | 0.60 ± 0.02 | 0.35 ± 0.04 |

DOI: https://doi.org/10.7554/eLife.39273.025

# Stabilization of codon-anticodon interaction by the ribosome, eIF5, eIF2α and eIF1A in py48S-eIF5N

The py48S-eIF5N is locked into a single configuration with the exception of small movements of eIF2 subunits γ, β, and domain 3 (D3) of the α subunit around the acceptor arm of the tRNA$_i$ (see maps

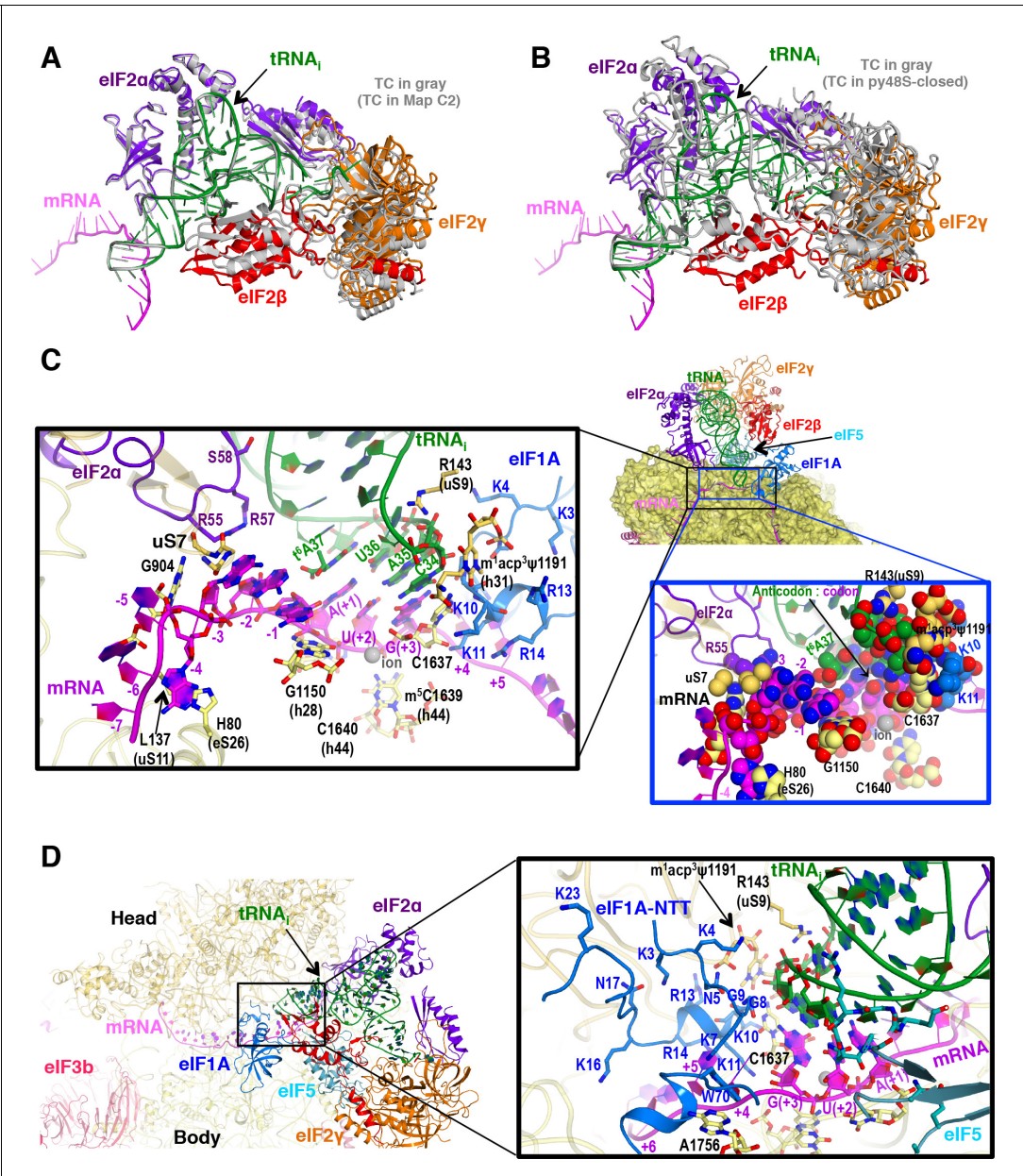

**Figure 6.** P site conformation and surrounding elements in py48S-eIF5N. (A) Cartoon representation of the TC in Maps C1 and C2, resulting from superposition of the 40S body in the two maps. The tRNA$_i$, mRNA, and each component of TC is colored differently for C1, whereas all components of C2 are in gray. (B) Cartoon representation of the TC in Map C1 and py48S-closed (PDB 3JAP), resulting from superposition of the 40S body in the two maps. The tRNA$_i$, mRNA, and each component of TC is colored differently for C1, whereas all components from py48S-closed are in gray. (C) Cross-section of the 40S subunit along the mRNA path and tRNA$_i$ bound in the P site of py48S-eIF5N, viewed from the top of the 40S subunit. eIF1A, eIF2 and eIF5-NTD are also shown. Black box inset: Detailed view of the codon-anticodon and surrounding elements that stabilize this interaction. Ribosomal, tRNA$_i$ and eIF2α residues involved in the interaction with mRNA at (minus) positions 5' of the AUG codon (+1) are also shown. Blue box inset: Spheres representation of the same region in (C) highlighting the close packing of mRNA from positions −4 to +3 with its stabilizing residues of the ribosome, tRNA$_i$ and eIF2α in the mRNA channel. (D) eIF1A NTT interactions with the codon-anticodon duplex and the 40S subunit in py48S-eIF5N.
DOI: https://doi.org/10.7554/eLife.39273.026

C1 and C2; *Figure 6A,B* and *Figure 1—figure supplement 1*). We observe an apparently greater accommodation of the codon-anticodon duplex in the P site (*Figure 2C*) in concert with stabilization of the ASL by other elements from the eIF5-NTD (noted above), the 40S body, the eIF1A-NTT and eIF2α-D3 (*Figure 6C*). In this regard, the ribosomal elements (R143 of 40S protein uS9 and rRNA bases from h44 and h31) involved in the stabilization of the codon-anticodon duplex at the P site described earlier for the py48S in the presence of eIF1 (*Hussain et al., 2014*) are also involved in py48S-eIF5N (*Figure 6C*). Moreover, a highly modified rRNA base at the 40S head (m$^1$acp$^3\psi$1191), not present in bacteria, plays a key role by providing a chair-like structure where the tip of the tRNA$_i$ ASL (C34) sits. Mutations in an enzyme involved in modifying this base are associated with a severe syndrome in humans (*Meyer et al., 2011*). Also, an ion (revealed as a spherical density) (*Figure 1—figure supplement 3D*) interacts with the phosphates of the U(+2) and G(+3) nucleotides of the AUG codon as well as the nearby rRNA residues G1150 and C1637, thereby playing a key structural role at the P site (*Figure 6C*). As in previous py48S structures in the P$_{IN}$ state (*Llácer et al., 2015*; *Hussain et al., 2014*), the eIF1A-NTT interacts with the codon-anticodon helix via Gly8-Gly9, which also allows the NTT to loop back towards the P site (*Figure 6D*). We are now able to visualize the entire NTT unambiguously except for the terminal Met1 residue. The NTT occupies the cleft in between the head and body of the 40S around the A and P sites, and several of its basic residues (Arg and Lys) establish contacts with rRNA residues from the 40S head and body (*Figure 6D*), essentially gluing them together to stabilize the closed conformation of the 48S. Recently, we established that substituting the conserved basic residues, as well as the yeast equivalents of eIF1A NTT residues identified as recurring substitutions in certain human uveal melanomas, decreases initiation at UUGs *in vivo* and selectively destabilizes PICs reconstituted at UUG codons in vitro (*Martin-Marcos et al., 2017*), as described above for eIF5-NTD hyperaccurate Ssu$^-$ substitutions.

## Interactions with the Kozak sequence of mRNA in py48S-eIF5N

In addition to the start codon, the mRNA bases at −1 to −4 in the E-site corresponding to the Kozak consensus sequence (*Kozak, 1986*) are locked into a single conformation in py48S-eIF5N. Bases −1 to −4 adopt an unusual but stable conformation, in which the adenine base at −4 is flipped out toward the 40S body, and the next three adenines (−3 to −1) stack with one another and are sandwiched by the uS7 β-hairpin loop and G1150 (from h28, at the neck of the 40S) (*Figure 6C*). Moreover, Arg55 of eIF2α interacts with the A nucleotide at −3, as previously reported (*Hussain et al., 2014*; *Pisarev et al., 2006*); and the t$^6$A37 base adjoining the tRNA$_i$ anticodon interacts with the A at −1 through its threonylcarbamyol modification and also stacks with the adenine base at the +1 position of the start codon (*Figure 6C*). Interestingly, the absence of t$^6$A37 in yeast increases translation initiation at upstream non-AUG codons (*Thiaville et al., 2016*), and therefore plays a role in stringent selection of AUG as start codon. In *S. cerevisiae,* A nucleotides at the −4 to −1 positions are highly preferred, particularly the A at −3 (*Zur and Tuller, 2013*), and are known to promote AUG recognition (*Martin-Marcos et al., 2011*; *Hinnebusch, 2017*). Placement of this favorable mRNA sequence at the E-site and its attendant interactions with elements of the 40S, eIF2α, and tRNA$_i$ might pause the ribosome during scanning and help position the downstream AUG codon at the P site. Its indirect stabilization of the codon-anticodon duplex and, therefore, of the closed conformation of the 48S might also facilitate dissociation of eIF1 after AUG recognition and subsequent eIF5-NTD binding.

## The path of mRNA and interaction with eIF3a at the mRNA exit channel of py48S-eIF5N

We could model the mRNA in the py48S-eIF5N from positions −14 to +17, spanning the entire mRNA channel plus additional nucleotides protruding from the two channel openings on the solvent side of the 40S (*Figure 7A*). The last four nucleotides at the 3' end and first 14 nucleotides at the 5' end of the mRNA could not be modeled owing to lack of unambiguous high-resolution density. As previously reported (*Hussain et al., 2014*), we observe kinks between the A and P codons and P and E codons (*Figure 6C*). At the mRNA entry site (*Figure 7B*), the latch is closed (*Figure 1—figure supplement 5*) and the mRNA interacts with both elements from the head (uS3) and body (uS5 and h16) of the 40S. We recently showed that two conserved Arg residues in yeast uS3 in proximity to the mRNA (R116-R117) stabilize PIC:mRNA interaction at the entry channel, augmenting this activity

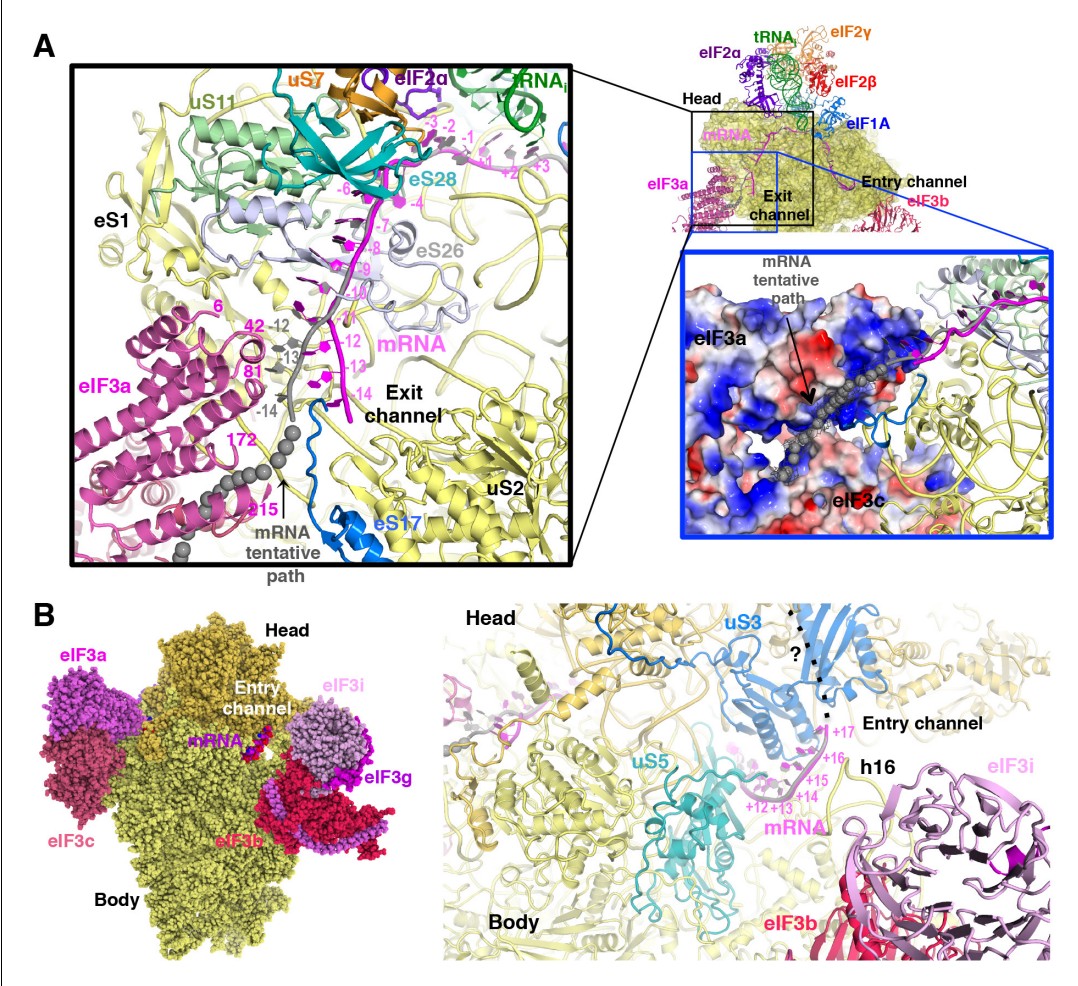

**Figure 7.** mRNA path at the exit and entry of the 40S mRNA channel in py48S-eIF5N. (**A**) Cross-section of the 40S subunit along the mRNA path of py48S-eIF5N, viewed from the top of the 40S subunit, showing both the entry and exit openings of the 40S mRNA channel. Path of the mRNA when the eIF3a/c PCI domains are present in high occupancy (i.e. in model A) is shown in gray. Black box inset: mRNA at the exit tunnel (from nucleotides −1 to −13). mRNA interacting proteins uS7, uS11, eS17, eS26 and eS28 are colored orange, light green, blue, blue-white and teal, respectively. The eIF3a PC1 domain is shown in pink, and a few residues in proximity to the mRNA path are labeled. Grey spheres highlight a tentative path for the mRNA based on an unassigned density in Map A (see tubular-shaped density shown as a grey mesh in the blue box inset). Blue box inset: Surface electrostatic potential of eIF3a/c PCI domains (blue: basic; red: acidic) supports their proposed interaction with mRNA upstream of position −11 depicted as the tubular-shaped density shown in grey mesh. (**B**) mRNA at the entry tunnel opening (from nucleotides + 11 to+17), colored as in A. mRNA-interacting 40S proteins uS3 and uS5 are colored blue and cyan, respectively. rRNA helix h16 also interacts with mRNA and is labeled. A possible trajectory for the 3' end of the mRNA is proposed and shown as a discontinuous black line.

DOI: https://doi.org/10.7554/eLife.39273.027

of eIF3, and are crucial for efficient initiation at UUG codons and AUG codons in poor Kozak context *in vivo* (*Dong et al., 2017*). At the 3' end, the mRNA does not protrude away from the ribosome but instead points upward and remains attached to the 40S head. Whether this reflects the limited length at the 3' end of the mRNA used here or represents the true trajectory of the mRNA 3' end remains to be determined. The fact that proteins like eIF3g and eIF4B, which can interact with both mRNA and proteins of the 40S head (*Cuchalová et al., 2010*; *Walker et al., 2013*), bind in this region may favor the latter hypothesis.

At the mRNA exit channel (*Figure 7A*), the mRNA interacts with uS7, uS11, eS17, eS26, eS28 and the 3' end of the rRNA. At the exit channel pore, the −12 to −14 nucleotides of mRNA interact with the eIF3a-PCI domain, and interestingly, this interaction seems to change the trajectory of the mRNA at its 5'end (*Figure 7A*, black inset), as seen in Map A (*Figure 1—figure supplement 1*)

containing higher eIF3 occupancy. In fact, an unassigned tubular density at the 5' end of the modeled mRNA (*Figure 7A*, blue inset), which may correspond to part of the unmodeled first 14 nucleotides of the mRNA, approaches and lays on the electropositive surface of eIF3a. This possible interaction would be consistent with previous experiments showing cross-linking of mammalian eIF3a to the −11 to −17 positions in mRNA (*Pisarev et al., 2008*), and that the eIF3a-PCI is critical for stabilizing mRNA binding at the exit channel (*Aitken et al., 2016*).

## eIF3 architecture in the py48S-eIF5N complex

The eIF3 subunits in py48S-eIF5N are located entirely at the 40S solvent side, as in mammalian 43S structures (*Bai et al., 2015*; *Hashem et al., 2013*) and the yeast 40S-eIF1-eIF1A-eIF3 complex (*Aylett et al., 2015*), except for the N-terminal helical bundle of eIF3c, which is still located at the subunit interface (*Figure 8A*) as observed previously (*Llácer et al., 2015*). The density of eIF3 in py48S-eIF5N does not allow 'de novo' modeling of atomic coordinates, but it does permit rigid-body fitting of previously reported structures of eIF3 subunits. The eIF3a-eIF3c PCI heterodimer sits near the mRNA exit tunnel, whereas the quaternary complex of eIF3b/eIF3i/eIF3g/eIF3a-Cter is found near the mRNA entry channel (*Figure 8A and B*) as in the aforementioned previous structures. The eIF3 submodules are roughly in the same locations in py48S-eIF5N compared to previous structures (*Aylett and Ban, 2017*; *Valášek et al., 2017*). The eIF3a-Cterm helix likely helps to position the eIF3b β-propeller and RRM domains as it runs beneath the eIF3b β-propeller and also interacts with eIF3b-RRM domain (*Figure 8B*).

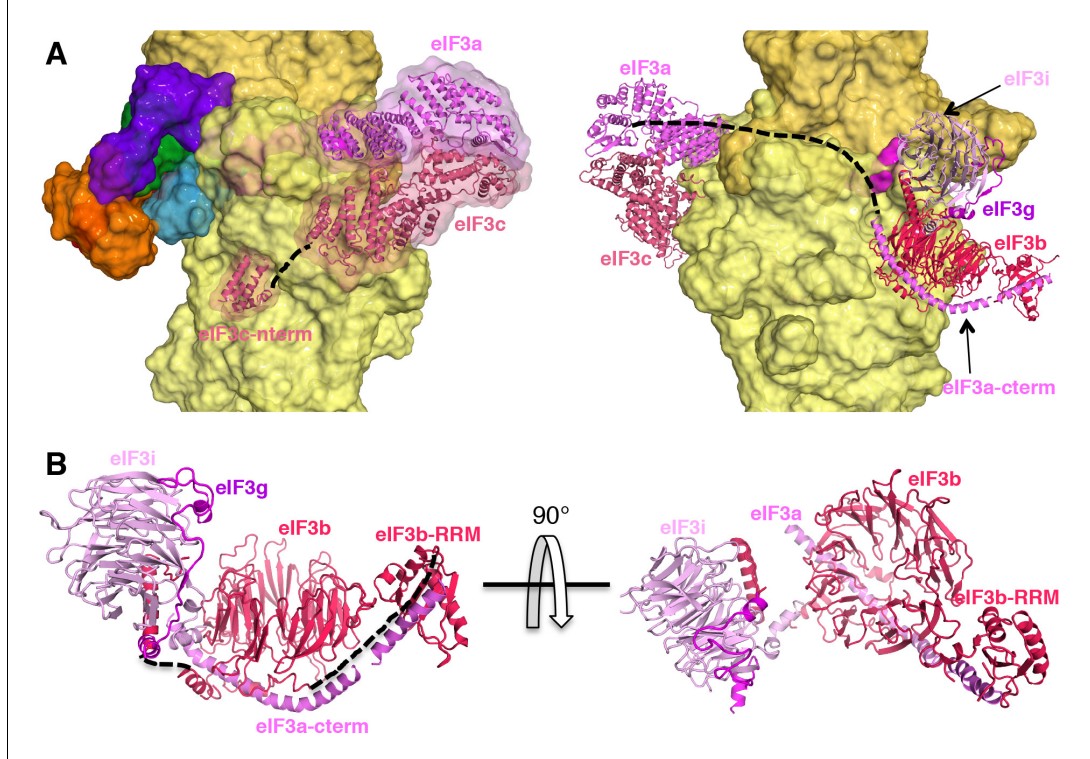

**Figure 8.** eIF3 architecture within py48S-eIF5N. (A) Two different views of the py48S-eIF5N PIC showing the locations of the different eIF3 subunits. All eIF3 domains, except for the eIF3c N-terminal helical bundle, reside on the solvent-exposed side of the 40S subunit. Shown as dashed black lines are the proposed linker connecting the eIF3c helical bundle and PCI domain; and the proposed path for the central part of eIF3a connecting the eIF3a/PCI domain and eIF3a C-term helix, of which the latter interacts with the eIF3b β-propeller. (B) Quaternary complex eIF3b/eIF3i/eIF3g/eIF3a-Cterm, shown in two different orientations. Unresolved connections between the eIF3b β-propeller and the eIF3b C-terminal helix and eIF3b RRM domain are shown as dashed black lines.

DOI: https://doi.org/10.7554/eLife.39273.028

## Discussion

In previous structures of yeast PICs containing Met-tRNA$_i$ base-paired to an AUG codon, the gate-keeper molecule eIF1 is still bound to the 40S platform, indicating that these structures likely depict intermediate states in the pathway prior to P$_i$ release from eIF2-GDP-P$_i$, which is gated by eIF1 dissociation. In the py48S and py48S-closed structures, where tRNA$_i$ is tightly enclosed in the P site, the location and conformation of the β-hairpin loops 1 and 2 of eIF1 are different from their counterparts in both the py48S-open complex and the simpler PIC containing only eIFs 1 and 1A. These changes to eIF1 are required to accommodate tRNA$_i$ binding in the conformation observed in the previously reported py48S closed complexes (*Llácer et al., 2015*; *Hussain et al., 2014*). The remodeling of eIF1 loop-1 disrupts certain interactions anchoring eIF1 to the 40S platform, presumably as a prelude to its eventual dissociation. Here, we describe a PIC representing a step further in the pathway, in which eIF1 has dissociated and is replaced by the eIF5-NTD on the 40S platform (*Video 1*). The eIF5-NTD interacts directly with the codon:anticodon duplex and might act to stabilize selection of an AUG start codon, as well as allowing greater accommodation of Met-tRNA$_i$ in the P site in a tilted conformation. The altered location and conformation of Met-tRNA$_i$ in the py48S-eIF5N complex is completely incompatible with eIF1 binding to the platform, even in the distorted state observed in the previous closed complexes. The eIF5-NTD, in contrast, is complementary to this new conformation and makes stabilizing interactions with the tRNA$_i$, thus promoting the fully closed/P$_{IN}$ state of the PIC and preventing rebinding of eIF1. The position and tilted conformation of Met-tRNA$_i$ in the py48S-eIF5N complex appears to set the stage for the next step in the initiation pathway, eIF5B-catalyzed subunit joining.

As shown previously (*Conte et al., 2006*), eIF5-NTD and eIF1 share the same protein fold, and they mostly contact the 40S subunit using analogous structural elements. Moreover, eIF1 and the structurally similar portion of eIF5-NTD contact essentially the same surface of the 40S platform. However, whereas eIF5-NTD binding is compatible with the more highly accommodated P$_{IN}$ state of Met-tRNA$_i$ and its tilting toward the 40S body observed in the py48S-eIF5N structure – and, in fact, appears to stabilize this fully accommodated state - eIF1 would clash extensively with Met-tRNA$_i$ in this location and orientation. The β-hairpin loops 1 and 2 of eIF5-NTD make extensive, favorable contacts with the Met-tRNA$_i$, and because eIF5 loop-2 is shorter/more basic and oriented away from the Met-tRNA$_i$, it avoids the electrostatic clash with the tRNA$_i$D-loop predicted for the larger/more acidic loop-2 of eIF1 that projects toward the tRNA$_i$ (*Figure 2C*). Supporting this view, we recently demonstrated that substituting acidic and bulky hydrophobic residues in eIF1 loop-2 with alanines or basic residues increases UUG initiation *in vivo* and stabilizes TC binding to the PIC at UUG codons, as would be expected from eliminating electrostatic/steric clashing, or introducing electrostatic attraction, between eIF1 loop-2 and tRNA$_i$, which removes an impediment to the P$_{IN}$ state to enhance selection of a near-cognate start codon (*Thakur and Hinnebusch, 2018*).

Our genetic findings provided evidence that electrostatic contacts of basic residues R28 and R73 in eIF5-NTD loops 1 and 2, respectively, with different segments of the tRNA$_i$ ASL stabilize the closed/P$_{IN}$ conformation at the start codon, as replacing them with Ala or Glu residues reduced initiation at UUG codons in yeast cells harboring the hypoaccurate Sui$^-$ variant eIF2β-S264Y. Consistent with this, the eIF5-R28A substitution also disfavored and destabilized the closed PIC conformation as judged by eIF1A dissociation kinetics, and decreased P$_i$ release from eIF2-GDP-P$_i$, primarily or exclusively at UUG codons in 48S PICs reconstituted with the eIF2β-S264Y variant. Similar findings were observed on introducing acidic residues at G29 and N30 in the eIF5-NTD loop-2. The relatively greater effects of these substitutions on UUG initiation *in vivo*, and in disfavoring/destabilizing the closed PIC conformation and reducing the rate of P$_i$ release in vitro at UUG versus AUG codons, taken in combination with earlier findings that the closed/P$_{IN}$ state of the PIC is inherently less stable at UUG versus AUG codons (*Maag et al., 2006*; *Nanda et al., 2009*), suggests that non-canonical 48S PICs at UUG codons are more sensitive to loop-2 mutations that disrupt electrostatic stabilization of the codon:anticodon duplex by the eIF5-NTD loop-2. Thus, our structural, genetic and biochemical evidence all suggest that contacts between the eIF5-NTD and the tRNA$_i$ ASL promote the closed/P$_{IN}$ conformation of the PIC, and thus are particularly important for efficient initiation not only at AUG codons but also at a near-cognate UUG codon, whereas eIF1 binding in virtually the same location on the 40S platform destabilizes the closed/P$_{IN}$ state and enforces a requirement for the perfect codon:anticodon duplex formed at AUG codons.

We found that introducing basic Lys or Arg residues at G29, N30, and E26, which should increase electrostatic attraction between eIF5-NTD loop-2 and the codon:anticodon helix, had the opposite effects of acidic substitutions at G29 and N30 at UUG codons, increasing the UUG:AUG initiation ratio *in vivo*, and favoring/stabilizing the closed PIC conformation and increasing the rate of $P_i$ release at UUG codons in vitro. These phenotypes mimicked those of the G31R substitution (*Maag et al., 2006*)—the first described Sui⁻ hypoaccuracy substitution in eIF5 encoded by *SUI5*. These results seem to indicate that increasing electrostatic attraction with loop-2 can stabilize mismatched codon:anticodon duplexes and thereby promote initiation at near-cognate start codons. However, G31R does not increase initiation at other near-cognate triplets besides UUG *in vivo* (*Huang et al., 1997*), and it disfavors the closed PIC conformation and slows down $P_i$ release at AUG codons in vitro (*Saini et al., 2014*) (*Figures 4* and *5*). Thus, the putative electrostatic stabilization seems to apply exclusively to the mismatched UUG:anticodon duplex and not those formed by other near-cognates, and to have the opposite effect on the perfect codon:anticodon helix formed at AUG. More work is required to understand the molecular basis of this exquisite specificity of basic loop-2 substitutions for UUG codons as well as the relative specificity for other non-canonical start codons.

The affinity of eIF1 for the free 40S subunit is ~30 fold higher than that of the eIF5-NTD (*Cheung et al., 2007*; *Nanda et al., 2013*), which would favor initial binding of eIF1 over eIF5. However, the affinity of eIF1 for the 43S·mRNA(AUG) complex is ~20 fold lower (*Cheung et al., 2007*), whereas the affinity of eIF5 for the same complex is >300 fold higher, in comparison to their respective affinities for the free 40S subunit (*Algire et al., 2005*). As a result, following AUG recognition, the affinity of eIF5 for the PIC exceeds that of eIF1 by two orders of magnitude. The relatively high rate of eIF1 dissociation from the PIC on AUG recognition (*Maag et al., 2005*) should allow the eIF5-NTD to compete with eIF1 for re-binding to the 40S platform, and the relatively higher affinity of eIF5 for the 43S·mRNA(AUG) complex should favor the replacement of eIF1 by eIF5-NTD on the platform.

The notion that the eIF5-NTD and eIF1 compete for binding to the PIC helps to explain previous findings in yeast (*Valásek et al., 2004*; *Nanda et al., 2009*; *Martin-Marcos et al., 2011*) and mammals (*Ivanov et al., 2010*; *Loughran et al., 2012*; *Terenin et al., 2016*) that overexpressing eIF1 or eIF5 have opposing effects on initiation accuracy, with overexpressed eIF1 increasing discrimination against near-cognate triplets or AUGs with sub-optimal Kozak sequences, and overexpressed eIF5 boosting utilization of poor start codons. This effect of overexpressed eIF1 implies that eIF1 dissociation from the PIC does not necessarily lead to an immediate release of $P_i$, an irreversible reaction; a fast rate of eIF1 re-binding driven by mass action can allow resumption of scanning and prevent $P_i$ release, consistent with the faster rate of eIF1 release than $P_i$ release from WT PICs reported previously (*Nanda et al., 2013*). By the same logic, overexpressed eIF5 would decrease reassociation of eIF1 through increased competition with the eIF5-NTD for binding to the 40S platform, and thus stimulate $P_i$ release at poor initiation sites. In addition to increasing competition by the eIF5-NTD with eIF1 for 40S-binding, eIF5 overexpression might also enhance the ability of the eIF5-CTD to more actively evict eIF1 from the 40S by competing for binding to the beta subunit of eIF2 (*Nanda et al., 2009*; *Llácer et al., 2015*).

The inference that $P_i$ release does not occur immediately on eIF1 dissociation is also supported by evidence of a functional interaction between the eIF5-NTD and eIF1A-CTT, involving movement of these domains towards one another in the PIC, which is required for rapid $P_i$ release on AUG recognition following the dissociation of eIF1 from the 40S subunit (*Nanda et al., 2013*). We previously speculated that movement of the eIF5-NTD toward eIF1A on AUG recognition would involve replacement of eIF1 with eIF5-NTD on the platform, based on the structural similarity between eIF1 and the eIF5-NTD and also evidence that eIF1 is in proximity to the eIF1A-CTT in the open, scanning conformation of the PIC but moves away from eIF1A on AUG recognition (*Maag et al., 2005*) at essentially the same rate that the eIF5-NTD and eIF1A-CTT move towards one another in the PIC (*Nanda et al., 2013*). The presence of the eIF5-NTD on the platform observed here in the closed py48S-eIF5N complex, in essentially the same location occupied by eIF1 in the 40S·eIF1·eIF1A and py48S-open complexes (*Llácer et al., 2015*), provides strong structural evidence supporting our previous hypothetical model.

The GAP activity of eIF5 is dependent on Arg15 at the N-terminal end of the NTD. Although Arg15 is in the vicinity of the GDPCP bound to eIF2γ in py48S-eIF5N, it is too distant to function in

stimulating GTP hydrolysis. Previous evidence indicates that GTP hydrolysis can occur in the scanning PIC and that $P_i$ release rather than hydrolysis per se is the step of the reaction most highly stimulated by AUG recognition. The position of eIF5-NTD in our structure is compatible with the notion that GTP hydrolysis has already occurred by the time the eIF5-NTD replaces eIF1 on the platform because the eIF5-NTD is engaged with the codon:anticodon helix rather than the GTP-binding pocket of eIF2γ (*Figure 2—figure supplement 1D*). Movement of the eIF5-NTD into the eIF1-binding site might be the trigger that releases $P_i$ from eIF2γ and could be the movement detected previously that brings the eIF5-NTD closer to the eIF1A-CTT upon start-codon recognition, a conformational change that is crucial for allowing $P_i$ release (*Nanda et al., 2013*). The presence of eIF1 on the 40S platform of the scanning PIC would potentially block access of the eIF5-NTD to the GTP molecule bound to eIF2γ. However, Arg15 resides within an extended NTT devoid of secondary structure that would likely be flexible enough to allow insertion of Arg15 into the eIF2γ GTP-binding pocket when the eIF5-NTD is tethered to the PIC via known interactions of the eIF5-CTD with the eIF3c NTD or eIF2β NTT (*Luna et al., 2012*).

The more accommodated position of Met-tRNA$_i$ and its tilting toward the 40S body observed with eIF5-NTD bound at the P site in py48S-eIF5N appears to set the stage for the binding of eIF5B and attendant subunit joining, as this tRNA$_i$ location/conformation is similar to that found in 80S initiation complexes with bound eIF5B (*Figure 2—figure supplement 3A*) (*Fernández et al., 2013*; *Yamamoto et al., 2014*). A recent study indicates that eIF5 and eIF5B cooperate in recognition of the start codon (*Pisareva and Pisarev, 2014*). The stabilizing effect of the eIF5-NTD on the more fully accommodated/tilted conformation of Met-tRNA$_i$ in the P site may form the basis of cooperation between eIF5 and eIF5B in recognition of AUG codons. In fact, the presence of the eIF5-NTD on the platform does not seem to impose any steric hindrance to the binding of eIF5B (*Figure 2—figure supplement 3B*). Interestingly, the tRNA$_i$ conformation in the py48S-eIF5N complex closely resembles that observed in bacterial translation initiation complexes containing IF2 (*Hussain et al., 2016*), the eIF5B homolog, suggesting that the processes of tRNA$_i$ accommodation in the P site converge on a similar position and orientation of tRNA$_i$ in the final stages of eukaryotic and bacterial translation initiation (*Figure 2—figure supplement 3A,C,D*). The mRNA path in the mRNA channel, mainly from −8 to +4 positions (*Figure 2—figure supplement 3F,G*), is also surprisingly similar between bacterial and eukaryotic initiation complexes.

Both major subdomains of eIF3 are visible in py48S-eIF5N almost exclusively on the solvent side of the 40S, as previously observed in mammalian 43S structures (*des Georges et al., 2015*; *Hashem et al., 2013*) and the yeast 40S-eIF1-eIF1A-eIF3 complex (*Aylett et al., 2015*). The eIF3a-eIF3c PCI domain is located at the mRNA exit tunnel and seen for the first time interacting with the −12 to −14 mRNA nucleotides, consistent with our recent biochemical analyses implicating the eIF3a-NTD in interaction with this region of the mRNA (*Aitken et al., 2016*). The quaternary complex of eIF3b/eIF3i/eIF3g/eIF3a-Cterm is located near the mRNA entry channel in the vicinity of the +12 to+17 mRNA nucleotides, which is also consistent with the same biochemical studies. Comparison of py48S-eIF5N to the previous py48S-open and py48S-closed structures containing eIF1 (*Llácer et al., 2015*) reveals a dramatic alteration of the position of eIF3b β-propeller, which moves from the entry channel on the solvent side in py48S-eIF5N to communicate directly with initiation factors on the interface surface of the 40S subunit in py48S-open and py48S-closed complexes. We originally interpreted the density map of py48S-closed as indicating binding of the eIF3i β-propeller in association with the 3g-NTD and 3b-CTD beneath the eIF2γ subunit of TC (*Llácer et al., 2015*). More recently it was suggested that the drum-like density attributed to the 3i β-propeller is actually the larger β-propeller of 3b, and that the density in contact with eIF1 is the 3b-RRM rather than the eIF3c-NTD (*Simonetti et al., 2016*). Based on a more recent, higher resolution map of a new py48S-open complex (*Llácer et al., 2018*); PDB 6GSM) we agree with these reassignments. The position of the eIF3b RRM at the interface surface of the py48S-open and -closed complexes, where it can contact eIF1, would clash with the eIF5-NTD bound to the platform in place of eIF1 in py48S-eIF5N (*Figure 9A–B*), which likely contributes to relocation of the eIF3b/eIF3i/eIF3g/eIF3a-Cterm module to the 40S solvent side, as observed here in the py48S-eIF5N complex.

As noted above, the solvent-side location of the eIF3b/eIF3i/eIF3g/eIF3a-Cter module has also been observed in yeast and mammalian PICs lacking mRNA, and we proposed previously (*Llácer et al., 2015*) that the transition of this module from the solvent side to the interface surface might be triggered by PIC attachment to the mRNA, and that its interactions with initiation factors

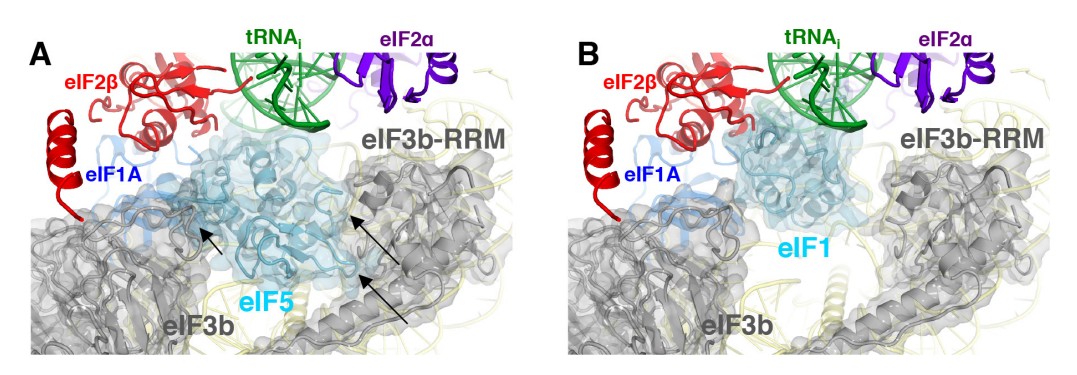

**Figure 9.** eIF3 relocates back to the solvent-exposed surface of 40S. (**A**) Modeling of the eIF3b/eIF3i/eIF3g/eIF3a-Cterm quaternary complex observed in py48S-closed-eIF3 (PDB 6GSN) at the subunit interface of py48S-eIF5N. The location of eIF5-NTD in the latter complex seems to be incompatible with this position of eIF3b/eIF3i/eIF3g/eIF3a-Cterm at the subunit interface, clashing with it at multiple points (highlighted with black arrows). (**B**) The eIF3b/eIF3i/eIF3g/eIF3a-Cterm quaternary complex in the py48S-closed-eIF3 complex (PDB 6GSN) shows no clashes with eIF1 bound at the P site.
DOI: https://doi.org/10.7554/eLife.39273.029

in the decoding center could help prevent dissociation of mRNA from the mRNA binding cleft during scanning. Thus, the view emerges that transition of the eIF3b/eIF3i/eIF3g/eIF3a-Cterm module from the solvent side to the interface side of the 40S occurs at the commencement of the scanning process and is reversed following AUG recognition and the replacement of eIF1 by the eIF5-NTD on the 40S platform.

Although conformational changes in eIF3 subunits may play a role in the transition between the open and closed states of the PIC, we did not previously observe an effect of eIF3 on the kinetics of eIF1A dissociation (*Maag et al., 2006*) or $P_i$ release (*Algire et al., 2005*) in the reconstituted system when PICs were assembled on the unstructured 43-mer model mRNA. It will be interesting in future studies to determine the effect of eIF3 and its subunits on these parameters with PICs assembled on natural mRNAs in the presence of the eIF4 factors.

The results of this study allow us to propose a model (*Figure 10*) of probable steps in translation initiation after the recognition of the start codon but prior to the subunit joining. Base pairing of Met-tRNA$_i$ with the AUG start codon in the $P_{IN}$ state of the closed conformation of the 48S PIC weakens eIF1 binding to the 40S subunit. Following eIF1 dissociation and attendant loss of its interaction with the eIF3b RRM, the eIF3b/eIF3i/eIF3g/eIF3a-Cter module relocates back to the solvent side of the 40S and allows binding of the eIF5-NTD at the site vacated by eIF1. The Met-tRNA$_i$ binds more deeply in the P site and tilts towards the 40S body in a manner stabilized by extensive interactions of eIF5-NTD loop-1/loop-2 residues with the AUG:anticodon duplex. The eIF5-NTD also prevents re-binding of eIF1 and a return to the scanning conformation of the PIC. As proposed previously (*Nanda et al., 2013*), the eIF5-NTD functionally interacts with the eIF1A-CTT to permit irreversible $P_i$ release and subsequent dissociation of eIF2-GDP. eIF5B is recruited via interaction with the extreme C-terminus of the eIF1A-CTT, captures the accommodated tRNA$_i$ in the P site and stimulates joining of the 60S subunit to produce the 80S initiation complex, leading to the final stage of initiation.

## Materials and methods

### Purification of ribosomes, mRNA, tRNA, and initiation factors

*Kluyveromyces lactis* 40S subunits were prepared as described earlier (*Fernández et al., 2014*). *Saccharomyces cerevisiae* eIF3 and eIF2 were expressed in yeast while eIF1, eIF1A, eIF5, eIF4A and eIF4B were expressed in *Escherichia coli* as recombinant proteins and purified as described (*Acker et al., 2007*; *Mitchell et al., 2010*). eIF4G1 was co-expressed with eIF4E in *E. coli* and both purified as a complex as previously described (*Mitchell et al., 2010*) with modifications. A Prescission cleavage site was introduced just before the eIF4G1 coding sequence on the reported

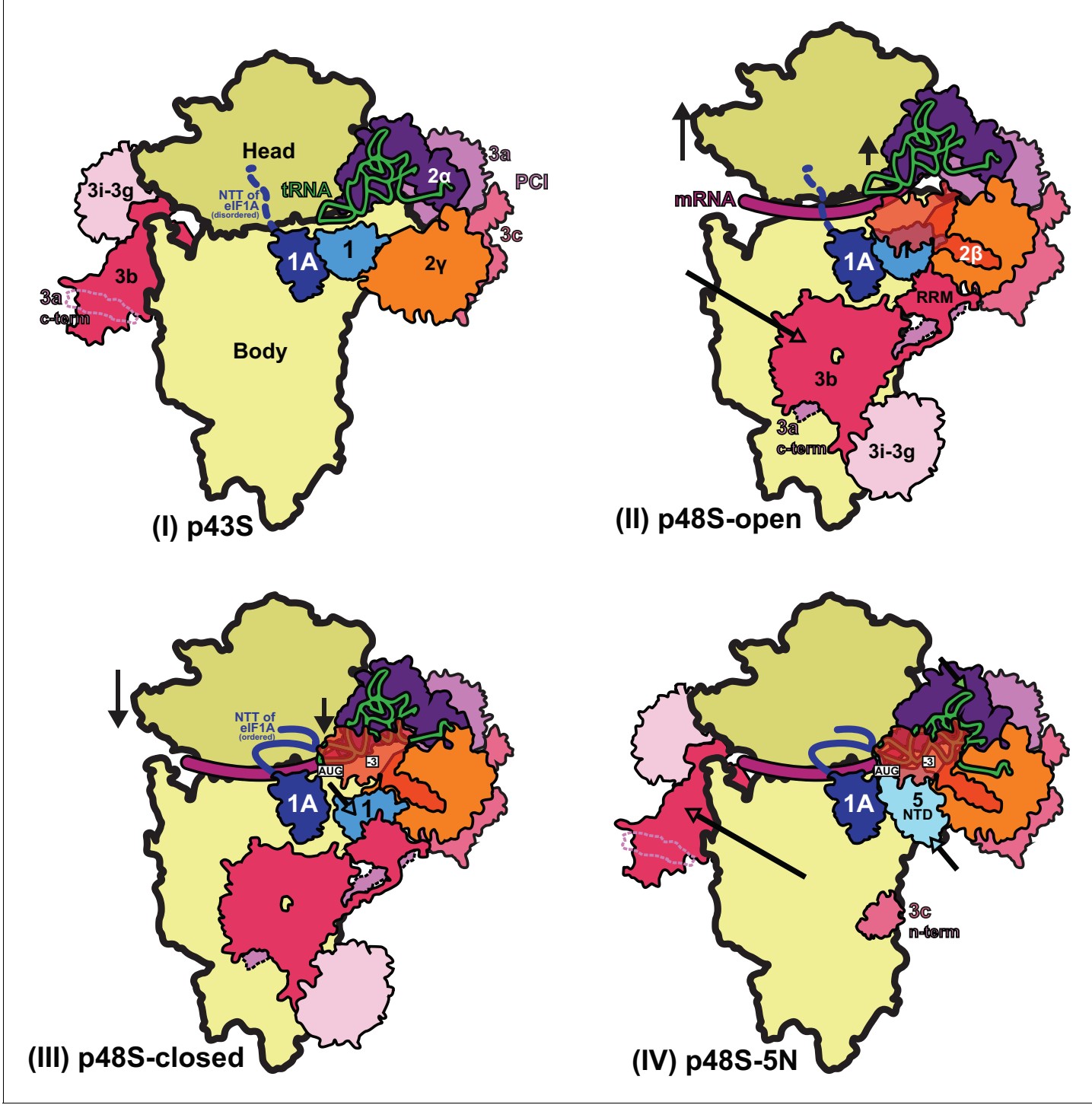

**Figure 10.** Schematic model of major conformational changes during initiation. (I) Binding of eIF1, eIF1A and eIF3 to the 40S subunit facilitates TC binding in the $P_{OUT}$ conformation to form the 43S PIC. The disordered NTT of eIF1A is shown as a dashed line. eIF2β and 3 c N-term were not resolved in this complex. (II) Upon mRNA recruitment facilitated by the upward movement of the 40S head (and also Met-tRNAi; shown by black arrows), which expands the mRNA entry channel, eIF3 undergoes major conformational changes and the eIF3b/eIF3i/eIF3g/eIF3a-Cterm module is repositioned from its initial position on the solvent exposed face of the 40S to the subunit-interface (shown by a pink arrow), contacting eIF2γ and eIF1. These contacts, together with eIF2β contacts with eIF1 and Met-tRNAi, probably stabilize this open scanning-competent conformation of the PIC (py48S-open). eIF2β is semitransparent overlapping eIF1.) (III) After recognition of the start codon, the 40S head moves downward (shown by a black arrow) to clamp in the mRNA and the ASL of the Met-tRNA$_i$ goes deeper into the P site ($P_{IN}$ state). eIF2β loses contact with eIF1 and Met-tRNAi and moves away to interact with the 40S head, the NTT of eIF1A stabilizes the codon:anticodon duplex, and eIF2α-D1 interacts with the −3 position of mRNA in the E site. eIF1 is

*Figure 10 continued on next page*

*Figure 10 continued*

also slightly displaced by Met-tRNA$_i$ from its original position on the 40S platform (shown by a cyan arrow). (IV) Base pairing of Met-tRNA$_i$ with the AUG start codon in the P$_{IN}$ state and Met-tRNA$_i$ tilting toward the 40S body (shown by a green arrow) weakens eIF1 binding to the 40S subunit causing eIF1 dissociation and, consequently, the eIF3b/eIF3i/eIF3g/eIF3a-Cter module relocates back to the solvent side of the 40S (shown by a pink arrow) and the eIF5-NTD binds at the site vacated by eIF1 (shown by a light cyan arrow) and stabilizes this Met-tRNA$_i$ conformation through extensive interactions of eIF5-NTD loop-1/loop-2 residues with the AUG:anticodon duplex. In subsequent steps (not shown), eIF5B binds and captures the accommodated Met-tRNA$_i$ in the P site and stimulates the joining of the 60S subunit to produce the 80S initiation complex, leading to the final stage of initiation.

DOI: https://doi.org/10.7554/eLife.39273.030

expression plasmid (*Mitchell et al., 2010*) in order to cleave and remove the GST tag from eIF4G1. Wild type tRNA$_i$ was overexpressed and purified from yeast and aminoacylated as described (*Acker et al., 2007*). The mRNA expression construct comprised a T7 promoter followed by the 49-nt unstructured mRNA sequence of 5'-GGG[CU]$_3$[UC]$_4$UAACUAUAAAA<u>AUG</u>[UC]$_2$UUC[UC]$_4$GAU-3' (with start codon underlined), cloned between XhoI and NcoI sites in a pEX-A2 plasmid (Eurofins Genomics). AUG context optimality was inferred from the sequences of highly expressed genes in yeast, as reported in *Zur and Tuller, 2013*. The mRNA was produced by T7 run-off transcription of the plasmid linearised by EcoRV (restriction site embedded in the mRNA sequence) according to a standard protocol. A 2 mL transcription reaction was resolved by electrophoresis on an 8M Urea, 12% acrylamide gel. A single mRNA band, visualized by UV light, was excised from the gel and mRNA was electro-eluted in TBE buffer, concentrated and buffer exchanged by dialysis into storage buffer (10 mM ammonium acetate, pH 5.0, 50 mM KCl). mRNA was capped with Vaccinia Capping System (New England Biolabs, M2080S) according to the manufacturer's instructions and further purified on an 8M Urea, 12% acrylamide gel as described above. The final concentration of mRNA was determined by A$_{260}$ measurement.

## Reconstitution of the 48S complex

First, a 43S mix was reconstituted by incubating 95 nM 40S subunits with eIF1, eIF1A, TC (consisting of eIF2, GDPCP and Met-tRNA$_i$), eIF3 and eIF5 in 40S:eIF1:eIF1A:TC:eIF3:eIF5 molar ratios of 1:2.5:2.5:2:2:2.5, in 20 mM MES, pH 6.5, 80 mM potassium acetate, 10 mM ammonium acetate, 5–8 mM magnesium acetate, 2 mM dithiothreitol (DTT), 1 µM zinc acetate. Separately, an mRNA-eIF4 complex was prepared, containing eIF4G1, eIF4E, eIF4A, eIF4B and capped mRNA in molar ratios of 1.5:1.5:5:2:2.5 with respect to the 40S ribosome in the final 48S mix (see below), in 20 mM Hepes, pH 7.4, 100 mM potassium chloride, 5 mM magnesium chloride, 2 mM DTT, 3 mM ATP). The volume of the mRNA-eIF4 mix was five times smaller than the 43S mix volume. Both the 43S mix and the mRNA-eIF4 mix were incubated separately for 5 min at room temperature before mixing them together to produce a 48S mix. After incubation for 2 min at room temperature, the sample (at a 40S final concentration of 80 nM) was cooled to 4°C and used immediately to make cryo-EM grids without further purification. When formaldehyde was used to crosslink the 48S complex (as described below), a solution at 3% in 48S mix buffer (at 1% final concentration of formaldehyde) was used just prior to making the cryo-EM grids.

## Electron microscopy

Three µl of the 48S complex was applied to glow-discharged Quantifoil R2/2 cryo-EM grids covered with continuous carbon (of ~50 Å thick) at 4°C and 100% ambient humidity. After 30 s incubation, the grids were blotted for 2.5–3 s and vitrified in liquid ethane using a Vitrobot Mk3 (FEI).

Automated data acquisition was done using the EPU software (FEI) on a Tecnai F30 Polara G2 microscope operated at 300 kV under low-dose conditions (35 e$^-$/Å$^2$) using a defocus range of 1.2–3.2 µm. Images of 1.1 s/exposure and 34 movie frames were recorded on a Falcon III direct electron detector (FEI) at a calibrated magnification of 104,478 (yielding a pixel size of 1.34 Å). Micrographs that showed noticeable signs of astigmatism or drift were discarded.

## Analysis and structure determination

The movie frames were aligned with MOTIONCORR (*Li et al., 2013*) for whole-image motion correction. Contrast transfer function parameters for the micrographs were estimated using Gctf

(*Zhang, 2016*). Particles were picked using RELION (*Scheres, 2012*). References for template-based particle picking (*Scheres, 2015*) were obtained from 2D class averages that were calculated from particles picked with EMAN2 (*Tang et al., 2007*) from a subset of the micrographs. 2D class averaging, 3D classification and refinements were done using RELION-1.4 (*Scheres, 2012*). Both movie processing (*Bai et al., 2013*) in RELION-1.4 and particle 'polishing' was performed for all selected particles for 3D refinement. Resolutions reported here are based on the gold-standard FSC = 0.143 criterion (*Scheres and Chen, 2012*). All maps were further processed for the modulation transfer function of the detector, and sharpened (*Rosenthal and Henderson, 2003*). Local resolution was estimated using Relion and ResMap (*Kucukelbir et al., 2014*).

A dataset of about 2100 images of non-crosslinked Sample 1 (*Figure 1—figure supplement 1*) was recorded from two independent data acquisition sessions. An initial reconstruction was made from all selected particles (*Aitken et al., 2016*) after 2D class averaging using the yeast 40S crystal structure (PDB: 4V88) low-pass filtered to 40 Å as an initial model. Next, a 3D classification into eight classes with fine angular sampling and local searches was performed to remove bad particles/empty 40S particles from the data. Two highly populated classes (70%; 276,269 particles) showed density for the TC. In a second round of 3D classification into four classes, only one of the classes (157,868 particles, 40% of the total; Map 1) still had clear density for the TC and corresponds to a closed conformation of the 48S with eIF5-NTD in place of eIF1, and yielded a resolution of 3.0 Å.

The map yielded a high overall resolution but poor local resolutions for peripheral and flexible elements like eIF3, eIF2β and eIF2γ, so we decided to collect an additional dataset in the presence of 1% formaldehyde. For this crosslinked dataset, about 1500 images were recorded using crosslinked Sample 2, and 312,041 particles were selected after two-dimensional classification. After obtaining an initial three-dimensional refined model, two consecutive rounds of 3D classification with fine angular sampling and local searches were performed. In the second round of 3D classification, only two classes containing TC were selected (113,838 particles, 36.5% of the total) and refined to high resolution after movie processing (3.6 Å). The model obtained is identical to that obtained with the non-crosslinked data although with higher occupancies for eIF3 and eIF2 (*Figure 1—figure supplement 6*).

Then the particles from both datasets (non-crosslinked Sample 1 and crosslinked Sample 2) were combined and we applied a strategy based on the reported method of masked 3D classifications with subtraction of the residual signal (*Bai et al., 2015*), by creating three different masks around the densities attributed to the PCI domains of eIF3, the eIF3bgi sub-module and the TC (*Figure 1—figure supplement 1*). We used 'focused' 3D classifications using these masks to isolate four distinct and well-defined maps, as follows.

A. Map A, showing higher occupancy and best local resolution for eIF3 PCI domains [53,870 particles, 3.5 Å],
B. Map B showing higher occupancy and best local resolution for eIF3 bgi sub-module [54,699 particles, 3.5 Å], and
C. By using the TC mask for focused 3D classifications we obtained the following two different maps showing a different TC conformation as a result of the rotation of subunits eIF2γ, eIF2β and domain D3 of eIF2α around the acceptor arm of the tRNA$_i$.
   1. Map C1 presents the TC in conformation 1 [74,772 particles, 3.5 Å],
   2. Map C2 presents the TC in conformation 2, and density for eIF2β is weaker [137,103 particles, 3.1 Å].

To ensure that the addition of the crosslinker to the latter dataset was not producing undesirable artifacts, each of these four classes was divided again into two subsets using the particles belonging to the crosslinked/non-crosslinked datasets and were refined independently; in all cases, the maps were identical to each other but at lower resolution than the one with all particles combined (see *Table 3*).

## Model building and refinement

In all five maps, the conformations of 40S, eIF1A, eIF5-NTD, tRNA, mRNA and domains D1 and D2 of eIF2α are identical. Thus, modeling of all these elements was done in the higher resolution map (3.0 Å, non-crosslinked dataset only; Map 1), whereas maps resulting from local masked classifications were essentially used for model building of the various subunits/domains of eIF3 and of eIF2β

and eIF2γ. In more detail, the atomic model of py48S in closed conformation (PDB: 3JAP) was placed into density by rigid-body fitting using Chimera (*Pettersen et al., 2004*). Then the body and head of the 40S were independently fitted. Wild type tRNA$_i$ was taken from PDB: 3JAQ for initial rigid-body fitting into its corresponding density. eIF3b subunit was taken from PDB: 4NOX and together with eIF3i and eIF3a C-term helix, placed into density by rigid-body fitting at the solvent side of the 40S in a position similar to that found previously (*Brown et al., 2015*). Possible residue numbering for eIF3a C-term helix is based on eIF3a secondary structure predictions and its known interactions with eIF3b and the eIF3b RRM domain (*Chiu et al., 2010*). Next, the NMR structure of the N-terminal domain of eIF5 (PDB: 2E9H) from *Homo sapiens* was docked into density using Chimera. Then, each chain of the model (including ribosomal proteins, rRNA segments, protein factors and tRNA and mRNA) was rigid-body fitted in Coot (*Emsley et al., 2010*) and further model building was also done in Coot v0.8.

Model refinement was carried out in Refmac v5.8 optimized for electron microscopy (*Brown et al., 2015*), using external restraints generated by ProSMART and LIBG (*Brown et al., 2015*). All maps, including the one of highest resolution (Map 1) and maps A, B, C1 and C2 were used for refining. Average FSC was monitored during refinement. The final model was validated using MolProbity (*Chen et al., 2010*). Cross-validation against overfitting was calculated as previously described (*Brown et al., 2015*; *Amunts et al., 2014*). Refinement statistics for the last refinements, done in Map 1, are given in *Table 1*. All figures were generated using PyMOL (*DeLano et al., 2006*) Coot or Chimera.

## Plasmid constructions

Plasmids used in the present study are listed in *Supplementary file 1* Plasmid pAS5-101 harboring *TIF5-FL* (*Saini et al., 2014*), encoding eIF5 tagged with FLAG epitope at the C-terminus, was used as template for constructing *TIF5* mutant alleles by fusion PCR. The resulting amplicons were inserted between the EcoRI and SalI sites of single-copy plasmid YCplac111, and the subcloned fragments of all mutant constructs were confirmed by DNA sequencing.

## Yeast strain construction

Yeast strains used in this study are listed in *Supplementary file 2* and were constructed by introducing the plasmid-borne *TIF5* mutant alleles into the *tif5Δ his4-301* strain ASY100 containing the *TIF5, URA3* plasmid p3342, which was subsequently evicted by counter-selection on medium containing 5-fluoro-orotic acid.

## Biochemical assays with yeast extracts

Assays of β-galactosidase activity in whole cell extracts (WCEs) were performed as described previously (*Moehle and Hinnebusch, 1991*). For Western analysis, WCEs were prepared by trichloroacetic acid extraction as described previously (*Reid and Schatz, 1982*) and immunoblot analysis was conducted as described (*Olsen et al., 2003*) using antibodies against FLAG epitope (Sigma #F-3165) to detect eIF5-FL proteins, or against eIF2Bε/Gcd6 as a loading control (*Bushman et al., 1993*).

## Assays of rates of eIF1A dissociation and P$_i$ release from reconstituted 43S·mRNA PICs

### Kinetics of eIF1A dissociation

The kinetics of eIF1A dissociation from reconstituted PICs were measured as described previously (*Saini et al., 2014*). The PIC was formed with 15 nM eIF1A-Fl (fluorescein-tagged eIF1A), 1 μM eIF1, 1 μM WT or mutant eIF5, 120 nM 40S subunits, 10 μM unstructured 43-mer model mRNA (GGAA(UC)$_7$UNUG(CU)$_{10}$C, where NUG is either AUG or UUG), 300 nM eIF2, 150 nM tRNA$_i$ and 0.5 mM GDPNP•Mg$^{2+}$ in 30 mM HEPES (pH 7.4), 100 mM potassium acetate, 3 mM Mg(OAc)$_2$, and 2 mM DTT. Reactions were incubated at 26°C for 45 min and fluorescence anisotropy (R$_{bound}$) was measured in Fluorolog-3 spectrofluorometer (Jobin Yvon Horiba). The fluorescein fluorophore was excited at 497 nm and fluorescence anisotropy was monitored at 520 nm. Dissociation of eIF1A-Fl was initiated by addition of a chase consisting of 100-fold excess (1 μM) unlabeled eIF1A and the change in fluorescence anisotropy of eIF1A-Fl was measured over time for up to 3 hr. Anisotropy

values were plotted as a function of time and data were fit with a double-exponential decay equation. The individual curves were than end-point normalized, based on the endpoints calculated from the raw curves. All experiments were performed at least in duplicate. The values of kinetic parameters calculated from the eIF1A dissociation curves has been presented as means ± average deviations.

### $P_i$ release kinetics

The kinetics of $P_i$ release from reconstituted 43S PICs in response to start-codon recognition was measured using a rapid-quench device (Kintek) as decribed previously (*Saini et al., 2014*). TC was formed at 4X concentration: 3.2 μM eIF2 or Sui3-2 eIF2, 3.2 μM tRNA$_i$ and 250 pM γ[$^{32}$P]-GTP in 1X recon buffer (30 mM HEPES•KOH, pH 7.6, 100 mM KOAc, 3 mM Mg(OAc)$_2$ and 2 mM DTT). 4X ribosomal complex was made using 800 nM 40S subunits, 3.2 μM eIF1, and 3.2 μM eIF1A. TC and ribosomal complex were mixed in equal volumes to make 2X PIC. Reactions were then initiated in a quench flow with an equal volume of a solution of 2 μM WT or mutant eIF5 and 20 μM model mRNA (AUG or UUG) in recon buffer and quenched at different times with 0.1 M EDTA. The samples were then run on PEI-cellulose TLC plates using 0.4 M KPO$_4$ buffer (pH 3.4) as the mobile phase. This was followed by phosphorimager analysis to quantify the fraction of GTP hydrolyzed over time. The data were fit with a double exponential rate equation. The first phase corresponds to GTP hydrolysis and the second-phase corresponds to to $P_i$ release that drives the reaction forward (*Algire et al., 2005*). All experiments were done at least in duplicate and the kinetic parameter for $P_i$ release are reported as means ± average deviations.

## Data resources

Five maps have been deposited in the EMDB with accession codes EMDB: 4328, EMDB: 4330, EMDB: 4331, EMDB: 4327, EMDB: 4329, for the sample one map, Map A, Map B, Map C1 and Map C2, respectively. Two atomic coordinate models have been deposited in the PDB with accession codes PDB: 6FYX, PDB: 6FYY, for models showing TC in conformation 1 and conformation 2, respectively.

## Acknowledgements

We are grateful to CG Savva, G McMullan for technical support with cryo-EM, T Darling and J Grimmett for help with computing and J Brasa for help with figures/movies. JL was supported by a FEBS postdoctoral fellowship. TH acknowledges start-up funds from the Indian Institute of Science. This work was funded by grants to from the UK Medical Research Council (MC_U105184332), Wellcome Trust Senior Investigator award (WT096570), the Agouron Institute and the Jeantet Foundation to VR; by the Department of Science and Technology, Government of India Grant [Int/NZ/P-2/13] to AKS; from the NIH (GM62128) formerly to JRL; the Human Frontiers in Science Program (RGP-0028/2009) to AGH, JRL and VR; and by the Intramural Research Program of the NIH (AGH, JRL).

## Additional information

### Competing interests

Alan G Hinnebusch: Reviewing editor, *eLife*. The other authors declare that no competing interests exist.

### Funding

| Funder | Grant reference number | Author |
| --- | --- | --- |
| Department of Science and Technology, Ministry of Science and Technology | Int/NZ/P-2/13 | Adesh K Saini |
| Human Frontier Science Program | RGP-0028/2009 | Alan G Hinnebusch |
| National Institutes of Health | GM62128 | Jon R Lorsch |

| Medical Research Council | MC_U105184332 | V Ramakrishnan |
|---|---|---|
| Wellcome Trust | WT096570 | V Ramakrishnan |
| Agouron Institute | | V Ramakrishnan |

The funders had no role in study design, data collection and interpretation, or the decision to submit the work for publication.

## Author contributions

Jose Luis Llácer, Conceptualization, Resources, Data curation, Formal analysis, Validation, Investigation, Visualization, Methodology, Writing—original draft, Writing—review and editing; Tanweer Hussain, Conceptualization, Resources, Data curation, Formal analysis, Investigation, Methodology, Writing—original draft, Writing—review and editing; Adesh K Saini, Formal analysis, Supervision, Funding acquisition, Investigation, Visualization, Writing—original draft, Writing—review and editing; Jagpreet Singh Nanda, Conceptualization, Resources, Formal analysis, Investigation, Visualization, Methodology, Writing—original draft, Writing—review and editing; Sukhvir Kaur, Yuliya Gordiyenko, Rakesh Kumar, Resources, Investigation; Alan G Hinnebusch, Conceptualization, Formal analysis, Funding acquisition, Writing—original draft, Writing—review and editing; Jon R Lorsch, Conceptualization, Formal analysis, Supervision, Funding acquisition, Writing—original draft, Writing—review and editing; V Ramakrishnan, Conceptualization, Supervision, Funding acquisition, Writing—original draft, Writing—review and editing

## Author ORCIDs

Jose Luis Llácer (iD) https://orcid.org/0000-0001-5304-1795
Adesh K Saini (iD) http://orcid.org/0000-0002-5445-5786
Jagpreet Singh Nanda (iD) https://orcid.org/0000-0001-5387-8683
Jon R Lorsch (iD) http://orcid.org/0000-0002-4521-4999

## Decision letter and Author response

Decision letter https://doi.org/10.7554/eLife.39273.049
Author response https://doi.org/10.7554/eLife.39273.050

# Additional files

## Supplementary files

• Supplementary file 1. Plasmids used in this study
DOI: https://doi.org/10.7554/eLife.39273.031

• Supplementary file 2. Yeast strains used in this study
DOI: https://doi.org/10.7554/eLife.39273.032

• Transparent reporting form
DOI: https://doi.org/10.7554/eLife.39273.033

## Data availability

Five maps have been deposited in the EMDB with accession codes EMDB: 4328, EMDB: 4330, EMDB: 4331, EMDB: 4327, EMDB: 4329, for the sample 1 map, Map A, Map B, Map C1 and Map C2, respectively. Two atomic coordinate models have been deposited in the PDB with accession codes PDB: 6FYX, PDB: 6FYY, for models showing TC in conformation 1 and conformation 2, respectively. All data generated or analysed during this study are included in the manuscript and supporting files. Source data files have been provided for Tables 4 and 5 and Figures 4 and 5

The following datasets were generated:

| Author(s) | Year | Dataset title | Dataset URL | Database and Identifier |
|---|---|---|---|---|
| Llacer JL, Hussain T, Gordiyenko Y, Ramakrishnan V | 2018 | Structure of a partial yeast 48S preinitiation complex with eIF5 N-terminal domain (Map A) | https://www.ebi.ac.uk/pdbe/entry/emdb/EMD-4330 | Electron Microscopy Data Bank, 4330 |

| | | | | |
|---|---|---|---|---|
| Llacer JL, Hussain T, Gordiyenko Y, Ramakrishnan V | 2018 | Structure of a partial yeast 48S preinitiation complex with eIF5 N-terminal domain (Sample Map 1) | https://www.ebi.ac.uk/pdbe/entry/emdb/EMD-4328 | Electron Microscopy Data Bank, 4328 |
| Llacer JL, Hussain T, Gordiyenko Y, Ramakrishnan V | 2018 | Structure of a partial yeast 48S preinitiation complex with eIF5 N-terminal domain (Map B) | https://www.ebi.ac.uk/pdbe/entry/emdb/EMD-4331 | Electron Microscopy Data Bank, 4331 |
| Llacer JL, Hussain T, Gordiyenko Y, Ramakrishnan V | 2018 | Structure of a partial yeast 48S preinitiation complex with eIF5 N-terminal domain (Map C1) | https://www.ebi.ac.uk/pdbe/entry/emdb/EMD-4327 | Electron Microscopy Data Bank, 4327 |
| Llacer JL, Hussain T, Gordiyenko Y, Ramakrishnan V | 2018 | Structure of a partial yeast 48S preinitiation complex with eIF5 N-terminal domain (Map C2) | https://www.ebi.ac.uk/pdbe/entry/emdb/EMD-4329 | Electron Microscopy Data Bank, 4329 |
| Llacer JL, Hussain T, Gordiyenko Y, Ramakrishnan V | 2018 | Date from: Translational initiation factor eIF5 replaces eIF1 on the 40S ribosomal subunit to promote start-codon recognition | http://www.rcsb.org/structure/6FYX | Protein Data Bank, 6FYX |
| Llacer JL, Hussain T, Gordiyenko Y, Ramakrishnan V | 2018 | Date from: Translational initiation factor eIF5 replaces eIF1 on the 40S ribosomal subunit to promote start-codon recognition | http://www.rcsb.org/structure/6FYY | Protein Data Bank, 6FYY |

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
