## [Decision Letter]

Thank you for submitting your article "Translational initiation factor eIF5 replaces eIF1 on the 40S ribosomal subunit to promote start-codon recognition" for consideration by *eLife*. Your article has been reviewed by Andrea Musacchio as the Senior Editor, a Reviewing Editor, and three reviewers. The reviewers have opted to remain anonymous.

The reviewers have discussed the reviews with one another and the Reviewing Editor has drafted this decision to help you prepare a revised submission.

We have received the comments from three experts in the field all of whom agree that this manuscript detailing the contributions of eIF5 to the accuracy of start codon selection in eukaryotes during the later stages of initiation has sufficient merit for publication in *eLife*. All three reviewers commented on the strength of the study in using a wide breadth of approaches to support their conclusions about the critical role of eIF5 in replacing eIF1 to stabilize AUG start codons (relative to non-optimal start codons). Importantly, all three commented on the claim that the non-hydrolyzable GTP analog GDPCP allowed the authors to visualize a post-GTP hydrolysis state – either this conclusion needs to be better supported or articulated, or a more complete coverage of the possibilities moved to the Discussion section of the manuscript. Finally, it was generally felt that the manuscript was overly long and somewhat hard to follow, this being in part a problem of terminology (Sui and Ssu, for example) and in part the need for more schematics detailing some of the experiments. Some broad suggestions for revision are included in the reviewer comments. In light of this broad agreement by the reviewers on the importance of this study, we recommend that this manuscript will be appropriate for publication in *eLife* once the reviewer comments are addressed.

*Reviewer #1:*

Llácer and colleagues report cryo-EM structures of pre-initiation 48S complexes, supplemented by biochemical and mutational analyses. This work shows that eIF5 binds in place of eIF1, next to Met-tRNA_i_, to stabilize an initiation complex on the AUG codon. β-hairpin 1 of eIF5 interacts with the codon-anticodon helix. Mutations of this region of eIF5 change the specificity toward the AUG codon and affect the equilibrium between the open and closed PIC on the AUG vs. the near-start UUG codon. These conclusions are made using a range of approaches, including mutant cell growth, monitoring dissociation of fluorescently labeled eIF1A, and monitoring the kinetics of phosphate release from eIF2. These approaches reveal similar results, strengthening the mechanistic implications of eIF5 made using the cryo-EM structures. In addition, the authors resolve and reassign the positions of several subunits of the elusive large eIF3 and describe their detailed interactions with both the 5' and 3' ends of the well-resolved mRNA fragment at the mRNA exit and entry tunnels, respectively.

Overall, this is a strong and well-written manuscript, which provides critical insights into the role of eIF5 in the mechanism of eukaryotic translation initiation. The following questions should be addressed in the revised manuscript:

1) The eIF5-bound structure is interpreted as "post-release", despite the presence of non-hydrolyzable GDPCP in eIF2. This interpretation is based on the position of the catalytic R15 and interaction of eIF5 with the codon-anticodon helix. But could this complex represent a pre-hydrolysis state? In this work, rates of dissociation of eIF1A are faster than or similar to those of P_i_ release – could this be consistent with the idea of this complex representing a pre-hydrolysis state? It would be good if authors discuss this possibility and suggest what would trigger the movement of the GTPase and R15 toward each other to catalyze GTP hydrolysis.

2) Kinetic data with fitted curves shown in Figure 4 and Figure 5, do not have error bars, whereas the legend for Figure 4 says "errors are calculated as average deviation". With some points falling off the curve, it is unlikely that two measurements for each point were nearly identical. Authors should add error bars or comment on their absence.

*Reviewer #2:*

In the present manuscript, the authors present evidence for how the accuracy of start codon selection in eukaryotes is achieved, in later stages of the translation initiation process. They use cryo-EM reconstructions, yeast genetics, and biophysical assays to identify how the N-terminus of eukaryotic initiation factor 5 (eIF5) replaces eIF1 to stabilize AUG start codons, as opposed to non-optimal start codons such as UUG. The high-resolution cryo-EM reconstructions of partial yeast 48S preinitiation complexes (PICs) reveal the positioning of the N-terminus of eIF5 next to the start codon-anticodon interaction, and movement of tRNA_i_ into a position more compatible with later steps of subunit joining mediated by eIF5B. This "closed" state (P_IN_) of the PIC helps explain a number of prior genetic and biochemical and biophysical observations that are augmented here by site-directed mutagenesis of two β-hairpin loops in eIF5 seen in the structure at high resolution.

Genetically, the authors use established genetic systems in *S. cerevisiae* that are either error-prone (Sui^-^) or hyperaccurate (Ssu^-^) in terms of start codon selection. They find that charged amino acids at the tips of the eIF5-NTD β-hairpins have important roles in modulating the accuracy of start codon selection, as indicated by the ratio of UUG:AUG start codon usage. Positively charged amino acids induce a more error-prone phenotype, whereas negatively charged amino acids cause start codon selection to become hyperaccurate.

The authors then use biochemical and biophysical assays to test how mutations in the eIF5-NTD affect eIF1 release from PICs, and rates of P_i_ release, which is thought to occur after start codon recognition and to be gated by eIF1 release. These assays provide complementary supporting evidence for the interpretations of the genetic experiments. They support the idea that favorable eIF5-NTD interactions with the mRNA:tRNA-anticodon stem loop (ASL) at the start codon drives PIC closure, from P_OUT_ to P_IN_ conformations, which then locks the PIC in a "committed" state of initiation.

Finally, the authors circle back to describe some additional features of the structure that they observe here, i.e. interactions along the mRNA and the Kozak sequence, and the positioning of parts of eIF3 in the present and prior structures.

Overall, this paper has potential to provide a compelling view of the late stage of initiation dependent on eIF5. Although it has been known that the eIF5-NTD has high homology to eIF1, the structures presented here, accompanied by genetic and biophysical/biochemical data, provides a mechanistic view of why these homologies matter in translation initiation.

That being said, the paper in its present form is overly long and hard to follow. It is too technically written in many parts, and the Discussion section is too long given the highlights of the paper. Some thoughts on how the paper could be improved are described below.

1) The authors use non-hydrolyzable GDPCP in their structural studies and see that the R15 from the eIF5-NTD is far from the bound nucleotide in eIF2. Yet, they jump to the conclusion that their structure represents the state of the 48S PIC after GTP hydrolysis (subsection “The eIF5-NTD replaces eIF1 on the 40S platform near the P site”). This is not at all clear from the structural results. Interpretation of this sort should be postponed until the Discussion section, where the authors should make their case more clearly.

2) Somewhere in the figures, the authors should provide extreme close-up views of the tips of the β-hairpins in eIF5-NTD, with the 3.0 Å density. Figure 1—figure supplement 3 is not adequate in this regard. Please include labels for the relevant amino acids.

3) The authors have to use composite maps and focused classification to identify peripheral elements of the PIC complexes, either due to low occupancy or structural disorder. For these reasons, it is not appropriate to spend too much effort or space interpreting these peripheral elements. For example, is the positioning of eIF3b/g/i observed due to the formaldehyde crosslinking of some of the particles? What about the positioning of the PCI regions of eIF3? It would be hard to say given how the data were merged. Additionally, the resolutions obtained in the present maps do not allow for the modeling required describing eIF3 in Figure 8 and Figure 9, and in the model in Figure 10. As part of their analysis, the authors cite Simonetti et al. (2016), however this paper does not report a bona fide initiation complex. As it is, the misinterpretation of ABCE1 as eIF3 was described in Mancera-Martínez et al. (2017) and in Heuer et al. (2017).

4) Instead of focusing on eIF3, the authors should show more of the mRNA path, i.e. provide zoom-ins for Figure 6 and Figure 7. This would be helpful given how much of the mRNA the authors see.

5) The genetic data are very hard to follow, due to the use of technical terms that will not be familiar to a general audience. The authors should define Sui^-^ as error-prone (or hypoaccurate, if they prefer that term) in terms of start codon selection whenever it is used, i.e. "error-prone Sui^-^ mutatations". Similarly, Ssu^-^ mutations should be described as hyperaccurate whenever the term is used. Although this may seem redundant, it would help readers follow the flow.

6) Related to point 5, it would be most helpful to have schematics showing how the genetic experiments work, as figure supplements.

7) For the biochemical/biophysical assays of P_i_ release and eIF1 release, the authors should provide schematics of how the assays work, and what the different components of the kinetic fitting report. For example, it is hard to know what the a_1_, a_2_, k_1_, k_2_, and R_bound_ parameters mean with respect to start codon selection without having to spend many times rereading the text.

8) The changes in R_bound_ are not meaningful without some sense of the errors in the measurements.

9) For the methods, unless Saini et al. (2014) is complete and doesn't refer to prior publications itself, the authors should provide the methods for the P_i_ and eIF1 release assays here.

*Reviewer #3:*

While it has been known for some time that eIF5 contributes to start site fidelity, the molecular mechanism to explain this phenomenon is poorly defined. To address this, the authors have used an impressive array of approaches (cryo-EM, biophysical assays, and genetics) to reveal how the N-terminal domain of eIF5 binds to the 40S subunit in place of eIF1 after its dissociation upon start site recognition. The binding of eIF5 to this position on the 40S promotes an altered Met-tRNA_i_ binding conformation, which ensures stringent start site selection. The structural model is appropriately tested using powerful biophysical and genetic assays. Overall, this is an important study that will move the field forward and I have only minor suggestions to help make some parts of the manuscript clearer to a broad audience.

It is stated that the position of eIF5-NTD likely represents a post P_i_ release state (subsection “The eIF5-NTD replaces eIF1 on the 40S platform near the P site”). This is based on the fact that Arg15 in eIF5-NTD, which is the catalytic residue for the GAP activity, is too far from eIF2 bound GDPCP to interact with. However, the genetic and biochemical assays indicate that the interaction between eIF5-NTD and the tRNA_i_-ASL is important for the fidelity of start codon recognition. If the eIF5-NTD-ASL interaction occurs after P_i_ release (in other word, after the start site has been selected), shouldn't the mutations affect start codon recognition no matter which type of codon invoked P_i_ release? The authors later discuss that movement of eIF5-NTD toward this binding site may trigger P_i_ release (Discussion section), which is perhaps more realistic. Perhaps the authors can address this apparent inconsistency in the main text to avoid confusion.

The authors cite their previous study (which then partially sites another previous study) to describe the in vitro biochemical assays in the Materials and methods. It is therefore not clearly explained what factors were actually included in these assays. For example, the authors state in Figure 4 that a 43S-mRNA complex is used, but it is not clear that eIF3 is actually in the complex. If eIF3 is not present (which seems to be the case in the authors' previous start site selection kinetic assays), would its presence alter the kinetics? The authors need to more clearly state what is or is not present in the biochemical assays (and clearly state limitations of the assays if eIF3 is indeed missing) to avoid any possible confusion.

---

## [Author Response]

Reviewer #1:[…]Overall, this is a strong and well-written manuscript, which provides critical insights into the role of eIF5 in the mechanism of eukaryotic translation initiation. The following questions should be addressed in the revised manuscript:1) The eIF5-bound structure is interpreted as "post-release", despite the presence of non-hydrolyzable GDPCP in eIF2. This interpretation is based on the position of the catalytic R15 and interaction of eIF5 with the codon-anticodon helix. But could this complex represent a pre-hydrolysis state? In this work, rates of dissociation of eIF1A are faster than or similar to those of P_i_ release – could this be consistent with the idea of this complex representing a pre-hydrolysis state? It would be good if authors discuss this possibility and suggest what would trigger the movement of the GTPase and R15 toward each other to catalyze GTP hydrolysis.

We do not think this is true, as eIF1A dissociation is extremely slow and does not occur on the time scale of AUG recognition and P_i_ release. Rather, the change in eIF1A dissociation rates is only a proxy for the alterations in eIF1A affinity that occurs between the open and closed states of the PIC.

GTP hydrolysis (without P_i_ release) is known to occur in the scanning PIC prior to AUG recognition, and the absence of eIF1 in the structure plus the more fully accommodated position of tRNA_i_ indicates that both scanning and AUG recognition have already occurred. Accordingly, GTP hydrolysis also would have already been stimulated by eIF5-R15. We believe that the GDPCP is mimicking the GDP-P_i_ state following GTP hydrolysis; however, P_i_ release cannot occur despite the absence of eIF1. We have removed the first mention of GDPCP and we have revised the sentence in subsection “The eIF5-NTD replaces eIF1 on the 40S platform near the P site” that may have been misleading the reader. The following sentence is also added to the aforementioned subsection to make our meaning clearer to the reader:

“Given that GTP hydrolysis occurs in the scanning complex but P_i_ release requires eIF1 dissociation (Algire et al., 2005), and noting that eIF1 is absent and replaced by eIF5-NTD, we presume that this complex represents a state following both GTP hydrolysis and eIF1 dissociation but that the use of non-hydrolyzable GDPCP has prevented P_i_ release”.

2) Kinetic data with fitted curves shown in Figure 4 and Figure 5, do not have error bars, whereas the legend for Figure 4 says "errors are calculated as average deviation". With some points falling off the curve, it is unlikely that two measurements for each point were nearly identical. Authors should add error bars or comment on their absence.

The kinetic curves shown in Figure 4 and Figure 5 are representative curves from at least 2 independent experiments. Table 4 and Table 5 show the average values of each measured kinetic parameter derived from the individual experiments. The errors are presented in the Tables and represent average deviations of the mean values. We have now clarified these points and thank the reviewer for bringing this inaccuracy to our attention.

Reviewer #2:[…]Overall, this paper has potential to provide a compelling view of the late stage of initiation dependent on eIF5. Although it has been known that the eIF5-NTD has high homology to eIF1, the structures presented here, accompanied by genetic and biophysical/biochemical data, provides a mechanistic view of why these homologies matter in translation initiation.That being said, the paper in its present form is overly long and hard to follow.

In the revised version we have shortened the manuscript and made modifications to make it easier to follow.

It is too technically written in many parts, and the Discussion section is too long given the highlights of the paper. Some thoughts on how the paper could be improved are described below.1) The authors use non-hydrolyzable GDPCP in their structural studies and see that the R15 from the eIF5-NTD is far from the bound nucleotide in eIF2. Yet, they jump to the conclusion that their structure represents the state of the 48S PIC after GTP hydrolysis (subsection “The eIF5-NTD replaces eIF1 on the 40S platform near the P site”). This is not at all clear from the structural results. Interpretation of this sort should be postponed until the Discussion section, where the authors should make their case more clearly.

We thank the reviewer for pointing this. We have answered this above (Comment 1 of reviewer 1). In short, we have removed a sentence and modified a sentence in subsection “The eIF5-NTD replaces eIF1 on the 40S platform near the P site”. These modifications should make our meaning clearer to the reader.

2) Somewhere in the figures, the authors should provide extreme close-up views of the tips of the β-hairpins in eIF5-NTD, with the 3.0 Å density. Figure 1—figure supplement 3 is not adequate in this regard. Please include labels for the relevant amino acids.

New figure (as Figure 2—figure supplement 2) showing density of hairpins of eIF5-NTD is now included.

3) The authors have to use composite maps and focused classification to identify peripheral elements of the PIC complexes, either due to low occupancy or structural disorder. For these reasons, it is not appropriate to spend too much effort or space interpreting these peripheral elements. For example, is the positioning of eIF3b/g/i observed due to the formaldehyde crosslinking of some of the particles? What about the positioning of the PCI regions of eIF3? It would be hard to say given how the data were merged.

We thank the reviewer for this suggestion. In the text, we present only the position of the eIF3 or peripheral elements in brief without going into the details. We have removed sentences about the conformational changes and Figure 8C-E from the manuscript.

The details of the data processing and how the selected particles from each data set were combined are presented in the Materials and methods section and Figure 1—figure supplement 1. We have also made a few modifications in the Materials and methods section to make our procedures clearer to the reader.

The maps obtained with and without crosslinker are almost identical and superimpose very well. We have included a new figure (Figure 1—figure supplement 6) in the manuscript to make this point. All eIF3 elements are observed without the use of crosslinker as well, and in identical positions. It is mentioned in Materials and methods section that crosslinker did not affect the conformation of eIF3 in the complex, but only increased the occupancy of eIF3.

Additionally, the resolutions obtained in the present maps do not allow for the modeling required describing eIF3 in Figure 8 and Figure 9, and in the model in Figure 10.

The present maps do not allow for ‘de novo’ modelling for atomic coordinates, as the referee has correctly pointed out. However, rigid body fitting of known eIF3 subunit structures is possible and we have done that. There is no ‘de novo’ modelling involved. We thank the reviewer for pointing this out, and we have removed figures (Figure 8C-E) and statements about conformational changes.

As part of their analysis, the authors cite Simonetti et al. (2016), however this paper does not report a bona fide initiation complex. As it is, the misinterpretation of ABCE1 as eIF3 was described in Mancera-Martínez et al. (2017) and in Heuer et al. (2017).

We thank the referee for this suggestion. We have removed the citation from the Introduction where initiation complexes are mentioned. However, we have kept the citation in the Discussion section where the position of the eIF3b WD domain on the subunit interface is mentioned.

4) Instead of focusing on eIF3, the authors should show more of the mRNA path, i.e. provide zoom-ins for Figure 6 and Figure 7. This would be helpful given how much of the mRNA the authors see.

Zoomed in view of the mRNA path is shown in Figure 6C (at P site), Figure 6D (at A site), Figure 7A (mRNA channel exit) and Figure 7B (mRNA channel entry) as insets.

We have made the insets in Figure 7A bigger and also added more labels for ribosomal components and residue numbers of eIF3a that are in proximity to the mRNA path.

5). The genetic data are very hard to follow, due to the use of technical terms that will not be familiar to a general audience. The authors should define Sui^-^ as error-prone (or hypoaccurate, if they prefer that term) in terms of start codon selection whenever it is used, i.e. "error-prone Sui^-^ mutatations". Similarly, Ssu^-^ mutations should be described as hyperaccurate whenever the term is used. Although this may seem redundant, it would help readers follow the flow.

The terms hypoaccurate/hypoaccuracy and hyperaccurate/hyperaccuracy have now been used in the text.

6) Related to point 5, it would be most helpful to have schematics showing how the genetic experiments work, as figure supplements.

A schematic figure (figure 3—figure supplement 1) explaining the genetic experiments is now included.

7) For the biochemical/biophysical assays of P_i_ release and eIF1 release, the authors should provide schematics of how the assays work, and what the different components of the kinetic fitting report. For example, it is hard to know what the a_1_, a_2_, k_1_, k_2_, and R_bound_ parameters mean with respect to start codon selection without having to spend many times rereading the text.

We have now added schematics of the eIF1A dissociation and P_i_ release assays in Figure 4 and Figure 5, as suggested. We have also added indicators in parentheses for the relevant values in Table 4 (open or closed) to indicate which state each corresponds to. In addition, we have added footnotes to the K_amp_ and R_bound_ columns to indicate what each corresponds to.

8) The changes in R_bound_ are not meaningful without some sense of the errors in the measurements.

Average deviations of R_bound_ values for two independent experiments have been shown as errors in Table 4A and Table 4B.

9) For the methods, unless Saini et al. (2014) is complete and doesn't refer to prior publications itself, the authors should provide the methods for the P_i_ and eIF1 release assays here.

The experiments were done under the same conditions with the same parameters as represented in Saini et al. (2014). However, for maximal transparency, we have now added the experimental details to the text, as suggested.

Reviewer #3:[…]Overall, this is an important study that will move the field forward and I have only minor suggestions to help make some parts of the manuscript clearer to a broad audience.It is stated that the position of eIF5-NTD likely represents a post P_i_ release state (subsection “The eIF5-NTD replaces eIF1 on the 40S platform near the P site”). This is based on the fact that Arg15 in eIF5-NTD, which is the catalytic residue for the GAP activity, is too far from eIF2 bound GDPCP to interact with. However, the genetic and biochemical assays indicate that the interaction between eIF5-NTD and the tRNA_i_-ASL is important for the fidelity of start codon recognition. If the eIF5-NTD-ASL interaction occurs after P_i_ release (in other word, after the start site has been selected), shouldn't the mutations affect start codon recognition no matter which type of codon invoked P_i_ release? The authors later discuss that movement of eIF5-NTD toward this binding site may trigger P_i_ release (Discussion section), which is perhaps more realistic. Perhaps the authors can address this apparent inconsistency in the main text to avoid confusion.

We thank the reviewer for pointing this. We have answered this point above (Comment 1 of reviewer 1). In short, we have removed a sentence and modified a sentence in subsection “The eIF5-NTD replaces eIF1 on the 40S platform near the P site”. These modifications should make our meaning clearer to the reader.

The authors cite their previous study (which then partially sites another previous study) to describe the in vitro biochemical assays in the Materials and methods. It is therefore not clearly explained what factors were actually included in these assays. For example, the authors state in Figure 4 that a 43S-mRNA complex is used, but it is not clear that eIF3 is actually in the complex. If eIF3 is not present (which seems to be the case in the authors' previous start site selection kinetic assays), would its presence alter the kinetics? The authors need to more clearly state what is or is not present in the biochemical assays (and clearly state limitations of the assays if eIF3 is indeed missing) to avoid any possible confusion.

Biochemical assays were done using an unstructured poly(UC) 43-mer mRNA that has a central AUG or UUG codon. We have previously shown that the presence of eIF3 in this reconstituted system with an unstructured model mRNA does not alter the rate of eIF1A dissociation (Maag et al., 2006) or P_i_ release (Algire et al., 2005)and thus eIF3 was not used in these assays. We have added text to the Discussion section making this caveat clear.